# Monocytic Differentiation in Acute Myeloid Leukemia Cells: Diagnostic Criteria, Biological Heterogeneity, Mitochondrial Metabolism, Resistance to and Induction by Targeted Therapies

**DOI:** 10.3390/ijms25126356

**Published:** 2024-06-08

**Authors:** Øystein Bruserud, Frode Selheim, Maria Hernandez-Valladares, Håkon Reikvam

**Affiliations:** 1Acute Leukemia Research Group, Department of Clinical Science, University of Bergen, 5007 Bergen, Norway; mariahv@ugr.es (M.H.-V.); hakon.reikvam@uib.no (H.R.); 2Section for Hematology, Department of Medicine, Haukeland University Hospital, 5009 Bergen, Norway; 3Proteomics Unit of University of Bergen (PROBE), University of Bergen, Jonas Lies vei 91, 5009 Bergen, Norway; frode.selheim@uib.no; 4Department of Physical Chemistry, University of Granada, Avenida de la Fuente Nueva S/N, 18071 Granada, Spain; 5Instituto de Investigación Biosanitaria ibs.GRANADA, 18012 Granada, Spain

**Keywords:** acute myeloid leukemia, prognosis, differentiation, monocytic, FAB classification, metabolism, apoptosis, proteomic, venetoclax, targeted therapy

## Abstract

We review the importance of monocytic differentiation and differentiation induction in non-APL (acute promyelocytic leukemia) variants of acute myeloid leukemia (AML), a malignancy characterized by proliferation of immature myeloid cells. Even though the cellular differentiation block is a fundamental characteristic, the AML cells can show limited signs of differentiation. According to the French–American–British (FAB-M4/M5 subset) and the World Health Organization (WHO) 2016 classifications, monocytic differentiation is characterized by morphological signs and the expression of specific molecular markers involved in cellular communication and adhesion. Furthermore, monocytic FAB-M4/M5 patients are heterogeneous with regards to cytogenetic and molecular genetic abnormalities, and monocytic differentiation does not have any major prognostic impact for these patients when receiving conventional intensive cytotoxic therapy. In contrast, FAB-M4/M5 patients have decreased susceptibility to the Bcl-2 inhibitor venetoclax, and this seems to be due to common molecular characteristics involving mitochondrial regulation of the cellular metabolism and survival, including decreased dependency on Bcl-2 compared to other AML patients. Thus, the susceptibility to Bcl-2 inhibition does not only depend on general resistance/susceptibility mechanisms known from conventional AML therapy but also specific mechanisms involving the molecular target itself or the molecular context of the target. AML cell differentiation status is also associated with susceptibility to other targeted therapies (e.g., CDK2/4/6 and bromodomain inhibition), and differentiation induction seems to be a part of the antileukemic effect for several targeted anti-AML therapies. Differentiation-associated molecular mechanisms may thus become important in the future implementation of targeted therapies in human AML.

## 1. Introduction

Acute myeloid leukemia (AML) is an aggressive malignancy characterized by the rapid expansion of immature transformed myeloid cells, i.e., leukemic cells with a differentiation block, high proliferative capacity and high antiapoptotic activity [1]. This is true for acute promyelocytic leukemia (APL) but also for the non-APL variants of the disease (referred to as AML in this article) that are discussed in this review.

Despite their differentiation, block human AML cells can show limited signs of early/initial differentiation, e.g., granulocytic, monocytic, erythroid or megakaryocytic signs of differentiation [2]. Furthermore, although morphological and/or molecular signs of AML cell differentiation were part of the previous WHO 2016 classifications (Table 1) [2], differentiation was not regarded as a major prognostic or predictive parameter in non-APL variants of AML [3]. In the present article, (i) we focus on monocytic differentiation, (ii) we review associations between differentiation and other AML cell characteristics and (iii) we discuss differentiation as a marker of susceptibility to, or a part of, the response to new targeted therapies.

## 2. Monocytic FAB-M4/M5 AML Cell Differentiation: Morphological and Functional Characteristics, Molecular Markers, Genetic Heterogeneity and Minimal Prognostic Impact for Patients Receiving Conventional Chemotherapy

### 2.1. Monocytic Differentiation According to the FAB (French/American/British) Classification

The FAB system was previously used for AML subclassification [2,5], and it should still be regarded as a standardized system to characterize and classify AML cell differentiation status. This classification was mainly based on morphology and was used also in the WHO 2016 classification for patients who did not fulfill the criteria for other subclasses. The WHO 2016 classification identified the main AML subclasses: (i) AML with recurrent genetic abnormalities, (ii) AML with myelodysplasia-related changes, (iii) therapy-related myeloid leukemia, (iv) myeloid sarcoma, (v) myeloid proliferation associated with Down syndrome and (vi) AML not otherwise specified [6]. The last main subset “not otherwise specified” was further subdivided based on morphological criteria of AML cell differentiation, and the two subsets: acute myelomonocytic and acute monoblastic and monocytic leukemia corresponded to the original FAB-M4/M5 classification, including AML cells with signs of monocytic differentiation [4]. Thus, the differentiation criteria used in the WHO 2016 classification for “AML not otherwise specified” was based on the FAB classification and still represent a standardized system for the description and classification of AML cell differentiation; the differentiation criteria (including monocytic differentiation) are described in detail, and these criteria can also be used to describe differentiation for AML in general (Table 1) [7,8]. The WHO 2016 criteria give detailed morphological descriptions, and histochemical characteristics together with a description of the most common surface markers, are also given in this WHO classification.

As stated above, monocytic differentiation according to the FAB classification mainly relies on the morphological recognition of monoblasts and promonocytes [9]. Monoblasts are large cells with abundant and moderate-to-intensive basophilic cytoplasm, and they may show pseudopods, fine azurophilic granules, vacuoles or Auer rods. The nucleus is usually round with lacy chromatin and one or more prominent nucleoli. Promonocytes have a more irregular/convoluted nucleus, less basophilic cytoplasm with vacuoles and sometimes more granulation. The characteristics of the FAB-M4/M5 subclasses are summarized in Table 1; a more detailed description of the molecular markers is given Appendix A.

### 2.2. The Genetic Heterogenety of Monocytic FAB-M4/M5 AML Patients

The FAB-M4/M5 patient subset is heterogeneous with regards to genetic abnormalities that include mutations associated with both adverse and favorable prognosis for patients receiving conventional intensive therapy based on cytotoxic drugs [4,9,10,11,12,13,14,15]. The following cytogenetic and molecular genetic abnormalities are frequently observed:Patients with FAB-M5 seem to have a relatively high frequency of poor-risk cytogenetics [4,9].Many patients with monocytic AML cells show normal or nonspecific cytogenetic abnormalities (e.g., +8 in FAB-M4 and t(8;16) in FAB-M5 [4]).The most common mutations observed for at least 10% of FAB-M4/M5 patients are *NPM1* (38%), *DNMT3A* (37%), *FLT3-ITD/TKD* (32%), *NRAS* (11%) and *RUNX* (10%) [9].Several patients with *NPM1* mutations fulfill the FAB criteria for monocytic differentiation, i.e., they can be classified as FAB-M4/M5 based on the morphological characteristics of their AML cells [14,15]. *NPM1* mutations are common both for FAB-M4 and FAB-M5 patients [11] and seem to be (one of) the most common mutations in FAB-M4/M5 AML [9]. Patients with *NPM1*-mutated chemotherapy-related secondary AML show similar survival as patients with de novo *NPM1*-mutated AML [16], and this is better survival than other patients with therapy-associated AML who have an adverse prognosis [17,18,19]. The fusion oncogene *NUP98:NSD1* shows a higher frequency in FAB-M4, and its detection at the time of first diagnosis is associated with an adverse prognosis [12].*RAS* mutations are significantly more frequent in FAB-M4/FAB-M5 and are associated with an increased risk of late relapse [13].*PTPN11* mutations are more frequent in adults with monocytic AML and are independently associated with a lower complete remission rate and overall survival [10].

Taken together, these examples illustrate the genetic heterogeneity of monocytic FAB-M4/M5 AML, including genetic abnormalities that are associated both with favorable and adverse prognoses.

It can be seen from Figure 1 and Appendix A that the molecular markers of FAB-M4/M5 differentiation are involved in the regulation of fundamental cellular functions, including (i) contact/communication with other cells in their common bone marrow microenvironment, especially endothelial cells that are important in the regulation of local angiogenesis and AML stem cell support in the stem cell niches, (ii) binding to and modulation of the extracellular matrix, (iii) intracellular signaling, including NF-κB-stimulated cytokine release, (iv) modulation of intracellular signaling and (v) receptor binding of extracellular mediators and endosomal/phagocytic functions. Despite these differentiation-associated modulations of fundamental cellular functions, the monocytic morphology does not have any major prognostic impact for patients receiving conventional intensive antileukemic therapy, including allogeneic stem cell transplantation (discussed in Section 2.6 below).

### 2.3. The Transcptomic Heterogeneity of Monocytic FAB-M4/M5 AML Cells

Monocytic AML is very heterogeneous with regards to their transcriptomic profiles, and this heterogeneity has been described in several studies:A previous transcriptomic study identified eight different AML subsets based on the transcriptomic profile, and monocytic FAB-M4/M5 cells were classified in four of these eight subsets [20].A smaller transcriptomic study included a subset of monocytic AML patients, and the main difference was an increased expression of cells involved in cellular adhesion/communication for the monocytic cells compared to the other subsets [21].A previous study was based on the expression of 93 monocyte-specific genes, and based on this expression profile for a group of AML patients, they could classify the patients into a monocytic (144 patients) and a majority of 381 non-monocytic patients [22]. However, the monocytic cluster included 118 FAB-M4/M5 patients, whereas the non-monocytic cluster included 92 FAB-M4/M5 patients.The heterogeneity was also observed in a study of 27 patients with FAB-M5 AML; these authors did mRNA profiling based on the expression of 85 genes encoding histone-modifying proteins [23].A transcriptomic analysis of leukemic cells classified as mixed-lineage acute leukemia showed that even a subset of these patients was characterized by the high expression of monocyte genes [24].

Taken together, these results show that monocytic FAB-M4/M5 cells are very heterogeneous with regards to their transcriptomic profile. It is difficult to explain the common differentiation characteristics by similarities in the transcriptional regulation, as common posttranscriptional regulatory mechanisms seem to be more likely.

### 2.4. Expression of Costimulatory Molecules and Checkpoint Ligands by AML Cells: High CD86 and PD-L1 Expression by FAB-M4/M5 AML Cells

Optimal T-cell activation requires costimulatory signals together with the signals initiated by recognition of the antigen peptide/major histocompatibility complex by the T-cell antigen receptor; such costimulatory signaling can be initiated by CD80 or CD86 molecules that bind to CD28 on the T cells [25,26]. Costimulatory molecules are also expressed by human AML cells; the leukemic cell expression of CD80 seems to be low for most patients, whereas CD86 expression is generally higher [25]. This study also described significantly higher CD86 expression for monocytic FAB-M4/M5 cells compared to other FAB subtypes [25], whereas this difference could not be observed in another study [26].

Immune checkpoints mediate immunosuppressive effects on T-cell responses; the best-studied checkpoint in AML is the T-cell expressed PD-1 (programmed cell death 1) and its ligand PD-L1 [27,28]. PD-1/PD-L1 blocking is explored as a possible cancer treatment that (i) inhibits effector T-cell proliferation/function, (ii) downregulates T-cell receptor expression and (iii) reverses antiapoptotic signals in the malignant cells (for references, see [27]). It may also induce a T-cell exhaustion phenotype [29] and promote the expansion of immunosuppressive regulatory T cells (Treg) [30,31]. The possible use of checkpoint inhibition in AML is supported by in vitro studies [31], animal studies [28,29,30] and an adverse prognostic impact of immune checkpoints expression [32].

Several studies have investigated PD-L1 expression in human AML cells [26,27,28,29,30,31,32,33,34,35,36,37,38,39,40,41,42,43,44,45,46]. Yang et al. [27] referred to a Chinese article describing increased PD-L1 expression in FAB-M4/M5, but this association has not been observed in other studies [25,26,35]. PD-L1 upregulation has also been described for *TP53-* [36,37], *FLT3-* and *NPM1* [38]-mutated AML cells, but it is not associated with the AML cell load [37,46]. An adverse prognostic impact of high PD-L1 expression for patients receiving intensive AML therapy has also been described [26,32,38,39]. Furthermore, PD-L1 is expressed by AML stem cells, and the expression is increased in relapsed/refractory AML [25,31,33,34]. Finally, PD-L1 ligation will also influence the regulation of AML cell proliferation and apoptosis through the activation of PI3K-Akt-mTOR signaling, as well as the modulation of glucose and fatty acid metabolism, in the AML cells [40,41]. Thus, high PD-L1 expression seems to be a part of a chemoresistant phenotype that shows no strong association with monocytic AML cell differentiation.

TIM3 (T-cell immunoglobulin and mucin domain-containing 3) is another checkpoint, and patients with chemoresistant AML have an increased frequency of PD-1 and TIM-3-positive T cells in their bone marrow [42]. This TIM-3/galectin-9 pathway seems to be involved in the immune escape of AML cells [43], and the dual targeting of PD-1/PD-L1 and TIM-3/galectin-9 may therefore be a possible therapeutic strategy in AML [44].

One should finally emphasize that AML cell expression of co-stimulators or checkpoint molecules can be altered by therapeutic interventions, e.g., modulation by myeloid/monocytic differentiation induced by all-trans retinoic acid (ATRA) or vitamin D, increased PD-1L expression after IFN-γ exposure (e.g., released during local T-cell activation) [45], increased PD-1L expression after TLR2/4 ligation [46], decreased PD-1L after the inhibition of mitogen-activated protein kinase 7 (also known as MEK) [46] and increased CD80 during decitabine treatment [47].

To conclude, the AML cell expression of costimulatory and inhibitory checkpoint molecules varies between patients but shows no strong associations with monocytic differentiation of the AML cells.

### 2.5. Increased Constitutive Extracellular Release of Soluble Mediators by Monocytic AML Cells

Primary human AML cells express several cytokine receptors. However, the profile of expressed cytokine receptors differs between patients, and monocytic AML cells seem to express significantly higher levels of receptors for the hematopoietic growth factors GM-CSF (granulocyte-macrophage colony-stimulating factor) and Flt3 ligand and lower levels of SCF (stem cell factor) receptors [48]. AML cells also show a constitutive release of a wide range of mediators, including CCL and CXCL chemokines, interleukins, hematopoietic growth factors, angioregulatory vascular endothelial growth factor (VEGF), proteases and protease inhibitors [49,50,51,52,53,54]. Several of these cytokines thus function as autocrine growth factors.

In a previous study, we demonstrated that AML patients could be divided into two main subsets based on their cytokine release profiles (mainly, interleukins and chemokines were examined); one subset showed generally high mediator release, whereas the other subset showed low release [49]. The high-release subset included several patients with the favorable inv16 abnormality and a significantly higher number of patients with monocytic AML cell differentiation (i.e., FAB-M4/M5), and these high-release patients had a significantly better survival after intensive conventional AML treatment [49]. A high release of proteases and protease inhibitors was also associated with monocytic differentiation [51]. Similar observations have also been made in other studies describing the increased release of CXCL8/IL8 [54], high expression of VEGF [52] and increased serum levels of IL2 [53] for patients with monocytic AML cells.

The constitutive release of CCL and CXCL chemokines has been studied more in detail [50]. These studies also identified a patient subset with generally low or undetectable chemokine release; the other patients showed generally higher chemokine levels but especially for CCL1-4, CXCL1 and CXCL8, and a smaller patient subset also showed relatively high levels of angioregulatory CCL5 and CXCL9-11. High chemokine levels seem to be caused by transcriptional upregulation involving NF-κB [50,54].

To conclude, monocytic FAB-M4/M5 cells show a high constitutive release of several chemokines responsible for leukocyte chemotaxis and/or the regulation of local angiogenesis (especially CCL5/CXCL8-11). Monocytic AML cells also show a high release of proteases and protease regulators, suggesting that these patients differ with regard to proteolytic modulation of the extracellular matrix and extracellular-soluble mediators. Finally, it should be emphasized that AML cells show the constitutive release of a wide range of other biologically active molecules (e.g., several extracellular matrix molecules, soluble adhesion molecules like L-selectin and ICAM1, apoptosis-regulating mediators, including the Fas ligand, and soluble forms of several cell surface receptors) [55,56,57], but it is not known whether the levels of these mediators are associated with monocytic AML cell differentiation.

### 2.6. The FAB Classification of AML Has No or Only Very Limited Prognostic Impact for Patients Receiving Intensive Conventional Antileukemic Chemotherapy

It can be seen from Figure 1 and Appendix A that the molecular markers of FAB-M4/M5 AML cell differentiation are involved in the regulation of fundamental cellular functions. However, despite these differentiation-associated characteristics, monocytic morphology does not have any major prognostic impact for patients receiving conventional intensive cytotoxic therapy possibly including allogeneic stem cell transplantation.

The possible impact of FAB/differentiation on survival for AML patients receiving allogeneic stem cell transplantation was investigated in a small study including 39 patients (median age 14 years) transplanted in the period of November 1976 to July 1983 [58]. These authors described an adverse prognostic impact of high peripheral blood leukocyte counts at the time of diagnosis (i.e., ≥20 × 10^9^/L; *p* = 0.001) and monocyte morphology (i.e., FAB 4/5, *p* = 0.05). Most patients in the FAB-M4/M5 group died from relapse, whereas most of the other patients died in remission, but the numbers were so small that a reliable statistical comparison was not possible in this study.

Another study investigated 5848 patients that met the morphological/molecular WHO 2016 criteria for AML not otherwise specified [15]. FAB M0 was then associated with a significantly lower complete remission rate and inferior relapse-free and overall survival when compared to FAB-M1/M2/M4/M5/M6, whereas the data for FAB-M7 were inconclusive. This significance was not observed when only analyzing patients without *NPM1* and *CEBPA* mutations; the authors therefore concluded that the FAB classification by itself did not provide prognostic information for these patients.

A third study was based on 1690 patients that were allotransplanted in the first complete remission [11]. These patients were also classified as having AML not otherwise specified according to the WHO 2016 classification. The authors described an association between FAB-M6/M7 and both decreased AML-free survival and increased non-relapse mortality. The posttransplant survival for these patients may then reflect not only an association between differentiation and susceptibility to pretransplant antileukemic cytotoxic treatment but the survival will also be influenced by posttransplant anti-AML immune reactivity, possibly reflecting differences in the expression of immune costimulatory/checkpoint molecules by the AML cells (see also Section 2.5) [59].

It is justified to conclude that the morphological signs of differentiation (i.e., the FAB classification), especially monocytic differentiation, do not have any major prognostic impact in AML, although the minority of patients with erythroid/megakaryocytic differentiation may represent an uncommon exception. This conclusion is justified, even though experimental studies have described FAB-associated differences with regard to susceptibility to pharmacological antileukemic effects, e.g., resistance to cytarabine/daunorubicin for FAB-M4/M5 patients [60].

### 2.7. The Role of Nonleukemic Bone Marrow Cells in AML: The Importance of Leukemia-Supporting Stromal Cells and the Prognostic Impact of Monocyte Infiltration

The bone marrow microenvironment includes the endosteal/osteoblastic stem cell niches with relatively low oxygen/nutrient levels and the most central perivascular niches enriched in oxygen and nutrients (for references, see [61]). Various nonleukemic cells contribute to these hematopoiesis-supporting niches, including osteoblasts, bone marrow stromal/stem cells, adipocytes and fibroblasts, as well as various immunocompetent cells, including monocytes/macrophages (see below) [62]. Competition for oxygen in the cell-rich AML bone marrow leads to expansion of the peripheral hypoxic area [61], and the AML cells seem to reeducate nonleukemic cells towards AML supporting phenotypes [61,63,64]. The stromal cells may also contribute to the recruitment/development of M2-like cancer-associated macrophages, and this process is cytokine-supported with regards to monocyte recruitment (e.g., CCL2 and CXCR2 ligands), differentiation (e.g., macrophage colony-stimulating factor (M-CSF)) and polarization (e.g., IL6 and CXCL8/IL8) [61]. Several of these mediators can be released both by various stromal cells, including AML-associated chemoprotective fibroblasts [61,65], and by the AML cells themselves, especially FAB-M4/M5 AML cells (Section 2.4) [49,50]. Finally, FAB-M4/M5 AML cells show a high expression of several chemokines involved in local T-cell chemotaxis [49,50]. Although effector T cells can mediate antileukemic effects (even though escape mechanisms may also exist) [66,67,68], activated T cells can also release cytokines that mediate direct (i.e., antiapoptotic hematopoietic growth factors), as well as indirect (e.g., recruitment/differentiation of M2-like macrophages), AML-supporting effects [69]. However, to the best of our knowledge, no previous studies have investigated whether the extracellular stroma or the composition/education of nonleukemic bone marrow cells differ between FAB-M4/M5 patients and other AML patients.

Some of the previous studies of bone marrow infiltrating immunocompetent cells in AML have been based on gene expression profiles of bone marrow samples and the detection of biomarkers or biomarker combinations associated with various immunocompetent cell subsets. One such study suggested that the AML bone marrow infiltration of immunocompetent cells is very heterogeneous and includes CD4^+^ T cells, neutrophils, monocytes/macrophages, dendritic cells, natural killer (NK) cells, myeloid-derived suppressor cells (MDSCs), Treg cells and immature B cells [70]. These authors also suggested that the bone marrow profile of immunocompetent cells differs between patients. Furthermore, they investigated the bone marrow expression of 42 genes regarded to be immunosuppressive, e.g., including upregulation of the checkpoint genes PD-1/*PDC1*, PD-L1/*CD274*, PD-L2/*PDCD1LG2*, *LAG3* and *CTLA4* (see Section 2.3). High expression of these genes was associated with the concomitant expression of unfavorable metabolic regulatory genes (see Section 3). Another bioinformatical study based on bone marrow gene expression profiles classified patients into favorable and adverse prognostic subsets based on their mitochondrial metabolic gene profile [71], and this study also suggested that high-risk patients showed a higher infiltration of immunocompetent cells and, especially, M2 macrophages, γδ T cells, MDSCs and Treg cells.

AML supporting bone marrow macrophages are increased both in preclinical models of AML and in AML patients, especially macrophages with a M2-like phenotype [61]. These M2 macrophages are characterized by the expression of CD115 (M-CSF receptor), CD163 and CD206 [61], and T-cell-derived cytokines (including IL4/IL10/IL13) contribute to their development [69,72], together with direct effects of the AML cells [64]. There seems to be a gradual switch from M1- to M2-like macrophages, but the bone marrow levels of M2 macrophages differ between patients, and high levels (defined as CD206-expressing cells) seem to be associated with decreased overall, as well as event-free, survival [61,73]. This last prognostic observation should be interpreted with great care, because CD206 is also expressed by immunosuppressive dendritic cells [74] and even by monocytic AML cells [75]. However, a similar adverse prognostic impact of M2-like macrophages was also described in another study using the alternative M2 marker CD163 [74,76] that is part of a more complex macrophage gene expression cluster [74].

To summarize, AML-supporting nonleukemic bone marrow cells constitute a heterogeneous population of mesenchymal/endothelial cells, as well as various immunocompetent cells. The composition of the immunocompetent cell varies between patients, and several studies suggest that macrophages and especially M2 macrophages (and possibly also growth factor-secreting activated T cells) mediate chemoresistance and have an adverse prognostic impact on patients receiving conventional intensive therapy [48,69]. Such supportive effects may be mediated by paracrine loops. However, the limited monocytic differentiation of the AML cells does not have a similar prognostic impact as the infiltration of nonleukemic monocyte subsets. Finally, additional studies are needed to clarify whether the high constitutive mediator release by FAB-M4/M5 AML cells educates nonleukemic stromal cells in a different way compared to other AML patients.

### 2.8. Concluding Comment: The Limited Monocytic AML Cell Differentiation Is Not as Sufficient to Cause a Similar Prognostic Impact as the Bone Marrow Infiltration of Normal Monocytes/Macrophages

Several molecular markers of monocytic AML cell differentiation are shared between normal monocytes/macrophages and monocytic AML cells (Figure 1), and these markers influence the fundamental biological functions of these leukemic cells. Monocytic AML cells also share other characteristics with normal monocytes/macrophages, e.g., the high release of a wide range of soluble mediators (Table 2). However, despite these similarities, the limited differentiation of the monocytic leukemic cells is not sufficient to reach a prognostic impact after conventional intensive antileukemic therapy, similar to the monocytic bone marrow infiltration (see Section 2.7). The explanation for this is probably that the macrophage-associated impact is caused by a specific subset, i.e., M2 macrophages.

## 3. Variations in the Mitochondrial Energy Metabolism of AML Cells: Possible Associations between Metabolic Differences and Prognosis after Conventional Intensive Therapy

As outlined above, mitochondria-related reprogramming is important in AML, and two recent studies therefore investigated the possible prognostic impact of mitochondria-related genes for patients receiving conventional intensive antileukemic therapy [70,71]. The first study was based on an analysis of 149 patients, and they constructed a prognostic model based on the expression/RNA levels of the following four metabolic genes (information from the Gene database, accessed on 6 October 2023) [70]:4-hydroxyphenylpyruvate dioxygenase like (*HPDL*, chromosome 1). This mitochondrial protein seems to function as a 4-hydroxyphenylpyruvate dioxygenase.Carnitine palmitoyltransferase 1A (*CPT1A*, chromosome 11). Mitochondrial oxidation of long-chain fatty acids is initiated by carnitine palmitoyltransferases and carnitine-dependent transport across the mitochondrial inner membrane.Isocitrate dehydrogenase (NAD(+)) 3 catalytic subunit alpha (*IDH3A*, chromosome 15). Isocitrate dehydrogenases catalyze the oxidative decarboxylation of isocitrate to 2-oxoglutarate. This NAD-dependent enzyme is localized to the mitochondrial matrix, and NAD-dependent isocitrate dehydrogenases catalyze the rate-limiting step of the tricarboxylic acid cycle. The encoded protein is a subunit of one of the NAD-dependent isocitrate dehydrogenases.Electron transfer flavoprotein subunit beta (*ETFB*, chromosome 19). This electron transfer flavoprotein shuttles electrons between primary flavoprotein dehydrogenases involved in mitochondrial fatty acid and amino acid catabolism and the membrane-bound electron transfer flavoprotein ubiquinone oxidoreductase.

The expression levels of these four genes in the first study [70] increased with the increasing cytogenetic risk stratification, although multivariate analyses suggested that this signature had an independent prognostic impact.

An alternative prognostic model also based on mitochondrial metabolic markers has been described [71]. Its development was based on several public transcriptome databases; 31 mitochondrial metabolism-related genes were first selected, and finally, a prognostic model was constructed based on five genes that differed from the four genes listed above [70] (information from the Gene database, accessed 6 October 2023):Enoyl-CoA hydratase, short chain 1 (*ECHS1*, chromosome 10). This mitochondrial matrix protein functions in the second step of the fatty acid-beta oxidation pathway.NADH:ubiquinone oxidoreductase subunit A1 (*NDUFA1*, chromosome X). This protein is an essential component of complex I of the respiratory chain that consists of at least 43 subunits, with 7 of them being encoded by the mitochondrial genome. This protein functions as an anchor to the inner mitochondrial membrane.NADH:ubiquinone oxidoreductase core subunit S2 (*NDUFS2*, chromosome 1). The protein is a core subunit of the mitochondrial respiratory chain complex I.Succinate dehydrogenase complex flavoprotein subunit A (*SDHA*, chromosome 5). This gene encodes a major catalytic subunit of succinate-ubiquinone oxidoreductase in complex III of the mitochondrial respiratory chain.Succinate-CoA ligase GDP/ADP-forming subunit alpha (*SUCLG1*, chromosome 2). This gene encodes a subunit of succinate coenzyme A ligase that catalyzes the conversion of succinyl CoA and ADP or GDP to a succinate and ATP or GTP.

AML patients could be subclassified into two subsets referred to as a high- and low-risk patients (i.e., with regards to the response to conventional therapy) based on the RNA expression of these five mitochondrial metabolism-related genes. The two patient subsets showed differential expression of 38 genes (i) involved in the regulation of cytokine production, (ii) being localized to endocytic/vacuolar/lysosomal membranes (iii) and being involved in peptide binding/antigen processing and presentation (all proteins are listed in Appendix A). The two main patient subsets also differed with regards to the regulation of DNA repair, and the high-risk subset included more elderly patients. Finally, the two patient subsets seemed to differ with regards to bone marrow infiltration of immunocompetent cells, i.e., high-risk patients showing generally higher infiltration and especially higher infiltration of M2 macrophages, γδ T cells, MDSC and T reg.

To conclude, these two studies show that mitochondrial proteins encoded by nuclear DNA are important for the sensitivity of AML cells to conventional intensive chemotherapy, and these key regulators are involved in glucose, fatty acid and amino acid metabolism [70,71]. Additional analyses in both studies suggested that high-risk patients differed from low-risk patients with regards to the infiltration of immunocompetent cells, and high mitochondrial-related gene risk scores were thus possibly associated with an immunosuppressive microenvironment [70,71].

## 4. Variations in the Mitochondrial Energy Metabolism of AML Cells: The Associations between Metabolic Differences and Monocytic FAB-M4/M5 AML Differentiation Is a Molecular Background for Resistance to Targeted Therapy

### 4.1. A Brief Overview of the Cellular Energy Metabolism

The cellular energy metabolism includes several pathways; some of them are briefly described below [77,78,79,80,81,82,83]:*Cytoplasmic glycolysis.* The glycolytic pathway starts with the formation of glucose-6-phosphate; this is followed by a multistep process, finally generating pyruvate that can be transported into mitochondria to the tricarboxylic acid cycle, be further metabolized to lactate or be used for the synthesis of alanine. The 3-phosphoglycerate formed in the glycolysis can also be used for amino acid synthesis. The conversion of glucose to lactate provides 2 ATP molecules per glucose molecule, whereas complete glucose oxidation can provide up to 36 ATP molecules.*Pentose phosphate pathway.* This alternative cytoplasmic pathway utilizes glucose-6-phosphate; the final products include NADPH and ribose-5-phosphate, which are used in the synthesis of nucleotides and nucleic acids. NADPH is an important reducing agent in many cellular reactions (including lipid synthesis), and it protects against oxidative stress by regenerating reduced glutathione [84].*The tricarboxylic acid cycle.* After mitochondrial entry, pyruvate can be metabolized to acetyl-coenzyme A (CoA) that enters the mitochondrial tricarboxylic acid cycle. Acetyl-CoA can also be derived from fatty acid and amino acid metabolism.*Citrate* from the tricarboxylic acid cycle can be used for cytoplasmic fatty acid synthesis.*The respiratory chain/oxidative phosphorylation.* The chemical energy from the tricarboxylic acid cycle is converted into a mitochondrial electrochemical gradient that consists of five molecular complexes called complex 1/NADH:ubiquinone oxidoreductase, complex II/succinate dehydrogenase, complex III/ubiquinol:cytochrome c oxidoreductase, complex IV/cytochrome c oxidase and the final complex V/ATP synthase that converts ADP to ATP. Complex II/succinate dehydrogenase is also part of the tricarboxylic acid cycle. All complex II components are encoded by nuclear DNA, whereas the other four complexes also include proteins encoded by the mitochondrial genome.*Lipid metabolism.* Mitochondrial fatty acid oxidation breaks down fatty acids to acetyl-CoA that can enter the tricarboxylic acid cycle, whereas fatty acid synthesis takes place in the cytosol [84].

Thus, there is a metabolic crosstalk between the mitochondria and cytoplasm and between glucose, lipid, nucleotide and amino acid metabolism. Several intracellular mediators, including mitochondrial members of the Bcl-2 family, function as dual regulators of both cellular metabolism and survival, and the balance between the various family members in the AML cells depends on the cellular differentiation (see Section 5 and Section 6).

### 4.2. AML Cells Show a Mitochondrial Dysfunction but Also Heterogeneity with Regards to the Regulation of Energy Metabolism: Increased Reactive Oxygen Species in Monocytic FAB-M4/M5 Cells

AML cells have an increased mitochondrial mass but also a mitochondrial dysfunction with reduced maximal respiratory capacity [85], and the induction of oxidative stress with increased levels of reactive oxygen species seems to have an antileukemic effect [85,86]. Furthermore, Bcl-2 targeting reduces oxidative phosphorylation in AML cells [87], inhibits the electron transport chain and decreases energy production but increases cellular reactive oxygen species and thereby decreases the glutathione levels [88]. This metabolic modulation seems important for the antileukemic effects of Bcl-2 inhibition in AML cells that depend on Bcl-2 for survival [88,89], but monocytic AML cells often do not have such a strong dependency on Bcl-2 [7].

Mitochondrial oxidative phosphorylation is a key cellular process to generate ATP (for detailed descriptions and additional references, see [81,82,90,91,92,93,94,95]). Furthermore, reactive oxygen species are reactive, unstable and reduced forms of oxygen derivatives that are metabolic by-products [96] formed especially by mitochondrial metabolism [90]. Reactive oxygen species promote genomic instability and can thereby contribute to leukemogenesis and chemoresistance [97,98,99], but they also modulate intracellular signaling through the oxidation of lysine residues of, for example, redox-sensitive transcription factors and enzymes [100]. The balance between the production of reactive oxygen species and antioxidant systems defines the cellular redox status; oxidative stress is then defined as an excess of reactive oxygen species [90,99].

Patients with FAB-M4/M5 AML variants have higher basal levels of reactive oxygen species than other AML patients [90]. This is not caused by associations between monocytic differentiation and mutations in electron transport chain proteins [91,92,93,94,95] (see Section 4.3); it is rather caused by transcriptional differences between monocytic and less differentiated AML cells due to upregulation of the genes involved in oxidative phosphorylation and thereby higher basal respiration [7]. However, these higher basal levels of reactive oxygen species are not associated with susceptibility/resistance to conventional intensive antileukemic treatment, i.e., the European Leukemia Net (ELN) prognostic stratification [90].

### 4.3. Mutations in the Electron Chain Component Genes: No Association with FAB-M4/M5

Mutations affecting the mitochondrial-encoded electron transport chain complexes are uncommon. One study described that mutations in complexes I, III, IV and/or V (ATP synthase) were present in only 8% of AML patients [91]. Mutations in complexes I and IV seemed to have an adverse prognostic impact. Furthermore, mutations in the NADH dehydrogenase subunit 4 (*ND4*) of complex I are also uncommon (<10%) [93,94], and at least adult patients with somatic mutations seem to have a favorable prognosis [93]. Finally, variations in the two mitochondrial genes encoding cytochrome c oxidase subunit I and II (*COI* and *COII*) have been investigated in 235 AML patients with normal karyotypes, and 37 patients with nonsynonymous variations in *COII* had an adverse prognosis [95]. Thus, these mutations are uncommon and seem to differ in their prognostic impact, but they show no association with monocytic differentiation and cannot explain the associations between monocytic differentiation and resistance for patients receiving targeted therapy.

## 5. Dual Functions of Mitochondrial Mediators: Their Regulation of Both Energy Metabolism and Cell Survival and Regulation of Mediator Expression by Differentiation

This section describes how several regulators of cell survival also regulate cellular/mitochondrial energy metabolism; the overview is based on observations in various cell types, and future studies have to clarify more in detail (i) which mechanisms are most important in human AML cells and (ii) how the levels/importance of the different mediators vary between the AML cells for various patient subsets, especially whether they depend on pretreatment leukemic cell differentiation or differentiation induction by targeted therapies (Table 3).

### 5.1. Mitochondrial Dynamics: Effects of Bcl-2 Family Members Depend on the Biological Context

Mitochondria rearrange through fission and fusion [100], and fragmentation is a part of the apoptotic process, and Bax, as well as Bak, can promote mitochondrial remodeling through mechanisms that seem independent of their effects on mitochondrial permeabilization and cytochrome C release [101]. The permeability effect of Bax/Bak can be modulated or antagonized by the other antiapoptotic Bcl-2 family members Bcl-2 or and Mcl-1 [101,102,103].

### 5.2. Cytochrome C Oxidase: A Proapoptotic Mediator Essential for Cellular Energy Metabolism

Cytochrome c oxidase is localized to the inner mitochondrial membrane; its metabolic role is to pass electrons from complex III to complex IV of the electron transport chain, and various death stimuli (including metabolic stress) converge on the triggering of cytochrome c release through the mitochondrial outer membrane permeabilization [104,105]. However, its proapoptotic activity also depends on its activation status; glucose-stimulated production of glutathione by the pentose phosphate pathway will keep cytochrome c in an inactive reduced state [104], whereas reactive oxygen species will oxidize it and thereby make it capable of caspase activation [104,105]. Thus, metabolic stress induces both proapoptotic release and the activation of cytochrome c [104].

### 5.3. Bcl-2: An Antiapoptotic Regulator Targeting the Metabolic Functions of Cytochrome C

Bcl-2 is an antiapoptotic protein that maintains the integrity of the mitochondrial membrane, regulates cytochrome c activity, prevents cytochrome c release, sequesters proapoptotic proteins (e.g., Bax and Bak) and increases mitochondrial respiration even during cellular stress to maintain a slight prooxidant state [106,107,108,109]. 

Cytochrome c oxidase is a molecular complex [106], and Bcl-2 regulates its activity through modulation of the balance between its Va and Vb subunits. Cellular stress (e.g., hypoxia and glucose deprivation) triggers Bcl-2 to preserve the Va subunit while depressing the Vb subunit, the final result being decreased cytochrome c activity and maintained levels of reactive oxygen species [110,111]. Thus, Bcl-2 seems to maintain an environment that is optimal for malignant cell survival by adjusting mitochondrial respiration to meet the energy requirements without increasing intracellular reactive oxygen species.

### 5.4. Mcl-1: An Antiapoptotic and Metabolic Regulator Controlled by Glucose Metabolism

Mcl-1 promotes ATP production and contributes to the maintenance of the mitochondrial membrane potential [103], but it is also under metabolic control and targeted for degradation in the absence of glucose [112,113]. The modulation of glucose metabolism may thus reduce/suppress Mcl-1 and thereby sensitize cells to apoptosis induction [114]. Furthermore, Mcl-1 (similar to Bcl-2) is frequently overexpressed in AML (including AML stem cells) and is then an important regulator of the tricarboxylic acid cycle, glycolysis and the pentose phosphate pathway [115].

### 5.5. Bcl-xL: An Antiapoptotic Mediator That Modulates Several Metabolic Steps

Bcl-xL expression reduces citrate levels and thereby the levels of cytosolic acetyl-CoA [116]; it stimulates oxidative phosphorylation but lowers mitochondrial oxygen consumption [117]. Furthermore, it is detected at the endoplasmic reticulum–mitochondrion interface and can then decrease the NAD/NADH ratio, increase the activity of the electron transport chain and alter the tricarboxylic acid cycle activity through modulation of the mitochondrial calcium levels [118].

### 5.6. The Proapoptotic Bcl-2 Family Members Bax, Bad, Puma and Noxa Modulate Cell Metabolism

Proapoptotic Bax and Bad do not only regulate mitochondrial dynamics (Section 4.1). (i) Bax regulates the level of reactive oxygen species [119], (ii) Bax and Bad together regulate endoplasmic calcium homeostasis [120], (iii) the proapoptotic effects of Bax seem to be at least partly mediated through the modulation of ceramide (i.e., lipid metabolism) [121,122] and (iv) Bad is regarded as a regulator of glucose metabolism [123].

Puma expression can be induced/increased by glucose deprivation [113]. Similarly, the Noxa levels are also decreased by the presence of glucose [124], but it is, in addition, regarded as a glycolysis-promoting mediator [124] and a regulator of the balance between glycolysis and the pentose phosphate pathway [77].

### 5.7. TP53: Maintenance of Mitochondrial Integrity and Multiple Effects on Metabolic Regulation

*TP53* is a tumor suppressor and a regulator of apoptosis and cell metabolism. Its metabolic effects include the suppression of glycolytic flux and upregulation of the tricarboxylic acid cycle through various mechanisms, including [84,125]:Crosstalk with the metabolic sensor mTORC1 (mammalian target of rapamycin complex 1).Suppression of glycolysis through downregulation of the glucose transporters, inhibition of glycolytic enzymes and indirect transcriptional effects. TP53 is also a modulator of the balance between the glycolytic and the pentose phosphate pathway, a downregulator of lipid synthesis, a driver of the oxidative phosphorylation and a maintainer of mitochondrial integrity.TP53 can reduce/eliminate the deleterious effects of oxidative stress by limiting the cellular damage or by using these species to eliminate cells damaged beyond repair.

It should be emphasized that these TP53 effects on cellular metabolism may be tissue-dependent and possibly also dependent on the type of cellular stress.

### 5.8. Sirtuins Represent Possible Links between Protein Acetylation and Mitochondrial Metabolism

Several sirtuin lysine deacetylases modulate cellular metabolism, e.g., lipid metabolism and the urea cycle [77]. Histone deacetylase inhibition is now regarded as a possible therapeutic strategy in human AML [126,127,128] (see also Section 7.2), because sirtuins can function as regulators of both cellular survival and metabolism in human AML cells [129,130]. On the other hand, protein acetylation is also influenced by the availability of the acetyl-CoA metabolite [77], and Bcl-xL reduces acetyl-CoA levels and thereby also protein acetylation [116]. Thus, acetyl-CoA couples the mitochondrial regulation of cell survival/metabolism and epigenetic regulation [77].

### 5.9. The Prognostic Impact of Glycolytic Profiles in AML Patients Receiving Intensive Therapy

Cytoplasmic glycolysis results in the formation of pyruvate that can be transported into mitochondria to become a substrate in further energy metabolism. A small study including 23 adult patients investigated the prognostic impact of the pretherapy level of glycolytic AML cell metabolism for patients receiving intensive therapy [131]. The authors described a favorable prognostic impact of high pretherapy glycolytic metabolism. Another small study including 19 patients suggested that high pretherapy mitochondrial respiration/oxidative phosphorylation was associated with chemoresistance and adverse prognosis [132]. These results have to be interpreted with great care due to the low number of examined patients, but they suggest that the balance between cytoplasmic glycolysis and mitochondrial oxidative phosphorylation reflects the chemosensitivity of human AML cells.

### 5.10. The Prognostic Impact of Mitochondrial Lipid Metabolism in AML Patients Receiving Intensive Conventional Chemotherapy

A fatty acid metabolic signature was recently constructed based on analysis of the RNA expression of nine selected genes involved in fatty acid metabolism [133]. These authors then identified two contrasting groups; patients with a low fatty acid risk score then had better survival, and this prognostic impact was independent of age, *FLT3* and *NPM1* mutations, leukocyte count and cytogenetic risk, both in the training and validation cohort. Furthermore, gene ontology analysis showed that the signature was dependent on mitochondrial metabolism, including the tricarboxylic acid cycle and oxidative phosphorylation. Finally, high scores were associated with FAB-M5 classification and intermediate/adverse karyotypes. Thus, the differentiation-associated regulation of lipid metabolism may be part of an adverse AML cell phenotype for patients receiving intensive conventional anti-AML therapy.

### 5.11. The Heterogeneity of Monocytic FAB-M4/M5 AML Cells: Differences in the Regulation of Oxidative Phosphorylation Are Associated with Differences in Mitochondrial Translation and the Risk of Relapse after Intensive Chemotherapy

Two previous studies have investigated the proteomic/phosphoproteomic profiles of monocytic FABM4/M5 AML cells at the first time of diagnosis. The first study investigated 26 patients with later AML relapse and 15 with long-term relapse-free survival after completed conventional intensive therapy [134]. When comparing relapse patients with undifferentiated (i.e., FAB-M1/M2) and monocytic FAB-M4/M5 pretreatment AML cells, a large number of proteins (911) and phosphosites (257) differed significantly between the two groups. Relapse patients with the myeloblastic FAB-M1/M2 phenotype showed higher levels of RNA-processing proteins and lower levels of the mediators involved in translation and vesicular trafficking. In contrast, FAB-M4/M5 relapse patients showed increased levels of the proteins involved in mitochondrial translation and oxidative phosphorylation. Importantly, a high abundance of proteins involved in mitochondrial translation and oxidative phosphorylation also distinguished FAB-M4/M5 patients with and without later relapse after intensive ALM therapy. These observations suggest that the molecular mechanisms behind the later development of relapse depend on the pretreatment differentiation status (including metabolic regulation) of the AML cells, and such differences also distinguish between FAB-M4/M5 patients with and without later relapse.

A second study compared FAB-M4/M5 patients with and without pretreatment *NPM1* mutations [135]. The *NPM1*-mutated patients showed increased levels of the proteins involved in the regulation of endocytosis/vesicle trafficking, whereas patients without this mutation showed increased levels, especially of the proteins involved in the regulation of cytoplasmic translation, including several ribosomal proteins. Thus, even though monocytic AML cell differentiation was common for these patients, their genetic heterogeneity and therefore a difference in the regulation of the fundamental cellular functions was reflected in their proteomic profiles.

## 6. Resistance to Bcl-2 Targeting Therapy Depends on Both Specific Molecular Mechanisms Associated with Monocytic AML Cell Differentiation, as Well as General Susceptibility Biomarkers Identified in Previous Studies of Conventional Anti-AML Therapy

### 6.1. Clinical Studies of Venetoclax-Based AML-Stabilizing Therapy: The Importance of Genetic Abnormalities Together with Monocytic Differentiation

The resistance of monocytic AML cells to therapeutic targeting of Bcl-2 with the selective inhibitor venetoclax is the best characterized example of differentiation-associated treatment resistance that also involves differences in metabolism (Table 4). However, it should be emphasized that the differentiation-associated resistance is an important, but not the only, mechanism of venetoclax resistance. The main observations with regards to monocytic differentiation and venetoclax resistance are summarized in Table 4 and are discussed in detail in this main section [136,137,138,139,140,141,142,143,144,145,146,147,148,149,150,151,152,153,154,155,156,157,158,159,160,161,162,163,164,165,166,167].

Several studies have investigated the antileukemic effects of combining venetoclax with hypomethylating therapy, especially azacitidine. The VIALE-A study [138] included 431 AML patients with previously untreated disease unfit for intensive chemotherapy due to comorbidity or age above 75 years. The population included 431 patients with a median age of 76 years, both for the 286 patients receiving azacitidine–venetoclax and the 145 patients receiving azacitidine–placebo. After a median follow-up of 20.5 months, the median overall survival was 14.7 months in the venetoclax group and 9.6 months in the control group; the complete hematological remission rate was also significantly higher for patients receiving azacitidine–venetoclax (0.37 vs. 17.9%, *p* < 0.001). Based on the results from this study, it can be concluded that the azacitidine–venetoclax combination was associated with prolonged survival, a higher remission rate and an acceptable toxicity comparable to the treatment with azacitidine alone.

Several studies have investigated the prognostic impact of genetic abnormalities for patients treated with hypomethylating agents (e.g., azacitidine) plus venetoclax:One study including 103 patients described high responsiveness associated with *ASXL1* mutations, whereas *FLT3-ITD* and *TP53* mutations were associated with decreased responsiveness [139,159].Another study of 81 patients suggested that high venetoclax sensitivity was associated with *NPM1* and *IDH2* mutations, whereas resistance was associated with activation of the signaling pathways dependent on *FLT3*, *RAS* or *TP53* [140].Adverse ELN classification has been associated with resistance, i.e., an adverse karyotype may also be important for susceptibility to venetoclax [139,141,159].Genetic clonal heterogeneity seems to be associated with venetoclax resistance [138,140].Venetoclax resistance is seen both for FAB-M4 and FAB-M5 AML [164].

These studies demonstrate that the monocytic differentiation-associated resistance to venetoclax-based therapy is not the only resistance mechanism; additional mechanisms associated with defined genetic abnormalities also contribute to the overall venetoclax resistance. These additional mechanisms of resistance/susceptibility associated with defined genetic abnormalities have also been identified for patients receiving intensive and potentially curative intensive conventional AML treatment, whereas differentiation is not associated with susceptibility/resistance for this conventional treatment. Thus, the FAB-M4/M5-associated resistance to venetoclax-based treatment is more specific and only one out of several mechanisms that contributes to the overall resistance/susceptibility, whereas the genetic-associated mechanisms have a more general effect and have a prognostic impact even for patients receiving conventional intensive chemotherapy.

A recent study summarized and analyzed all the available real-world studies of venetoclax in combination with hypomethylating agents as a first-line AML treatment for patients unfit for intensive therapy [142]. The pooled overall data for these studies also showed improved survival for patients receiving the venetoclax–azacitidine combination similar to the VIALE-A study [138,140]. These observations further illustrate the importance of differentiation for the responsiveness to this treatment.

### 6.2. Venetoclax Combined with Intensive Chemotherapy

The association between monocytic differentiation and resistance has been described for the disease-stabilizing combination of venetoclax with hypomethylating agents (see Section 6.2). The initial clinical studies show that a combination of venetoclax with intensive and possible curative antileukemic treatment is also feasible [143,144], but it is not known whether venetoclax causes any differentiation-dependent improvement of AML-free survival after such intensive therapy.

### 6.3. Venetoclax for the Treatment of Relapse after Allotransplantation

Furthermore, three small studies have investigated the use of venetoclax plus hypomethylating agents for the treatment of AML relapse after allogeneic stem cell transplantation [145,146,147]; this strategy is also feasible, but it is still not known whether venetoclax has any differentiation-associated clinical effects in this high-risk patient subset.

### 6.4. AML Cell Differentiation, Energy Metabolism and Venetoclax Resistance

Several experimental studies suggest that AML cell differentiation is important for the sensitivity to venetoclax and venetoclax-based antileukemic combination therapy:Ex vivo *drug responses.* A small ex vivo drug screening study including 34 patients described a lower antileukemic effect of venetoclax for FAB-M4/M5 monocytic AML cells than for undifferentiated FAB-M0/M1 AML cells [8]. Resistance was then associated with a low Bcl-2/Mcl-1 ratio. Another experimental study also described an association between myelomonocytic differentiation and venetoclax resistance [148]. These last authors also described that myelomonocytic leukemia and upregulated Bcl-2A1/CLEC7A, as well as mutations of *PTPN11* and *KRAS,* conferred resistance to venetoclax and venetoclax combinations. Finally, ribosomal protein S6 kinase alpha-1 (RPS6KA1) also seems to be involved in the development of venetoclax resistance in monocytic AML cells [150]. These ex vivo observation further support and extend the observations in clinical studies (see Table 1, Section 6.1).*Monocytic AML stem cells.* A recent study characterized AML stem cell subsets [158]. These authors described a patient heterogeneity; some patients only had primitive AML stem cells, other patients had monocytic stem cells characterized by the expression of monocytic protein/differentiation markers and monocytic-associated transcription factors and the third group had a mixture of these two stem cell phenotypes that, at least for some patients, corresponded to underlying mutational variations. The ability of the monocytic stem cells to recapitulate AML development was verified in patient xenograft models. Although relatively few patients were examined, the presence of monocytic AML stem cells before venetoclax/azacitidine and/or the development of disease progression with monocytic AML cells during treatment were associated with a shorter duration of responses. The venetoclax/azacitidine combination then seemed to inhibit the electron transport chain and thereby cause the in vivo selection of monocytic AML cells, including monocytic AML stem cells [7,88].*Relapsed monocytic AML cells rely on Mcl-1 for metabolic regulation and survival.* Studies of the total AML cell population have shown that the venetoclax/azacitidine combination decreases oxidative phosphorylation and energy production, inhibits the electron transport chain, increases cellular reactive oxygen species and decreases glutathione levels [88,89]. Clinical venetoclax resistance can then be due to relatively low Bcl-2 levels as part of monocytic differentiation, and relapsed monocytic cells instead seem to rely on Mcl-1 for oxidative phosphorylation and survival [7,8,88].*Reactive oxygen species in FAB-M4/M5 AML.* Patients with the FAB-M4/M5 variants of AML have higher basal levels of reactive oxygen species (see Section 4.2) [90]. The monocytic AML cells seem to be transcriptionally distinct from undifferentiated/primitive AML cells; they also show an upregulation of genes important for oxidative phosphorylation and a higher basal respiration rate [7]. However, further studies are needed to clarify whether or how/whether these metabolic mechanisms contribute to monocytic-associated venetoclax resistance.*Patients with venetoclax resistance show a genetic heterogeneity.* Various genetic abnormalities are associated with the susceptibility of AML cells to venetoclax (Section 6.1). First, venetoclax-based treatment is highly effective in *NPM1*-mutated AML, but it is not known whether this is true also for monocytic *NPM1-Ins* AML cells [14,140,152,153]. Second, *PTPN11* and *KRAS* mutations are associated both with monocytic AML cell differentiation [13,154,155] and venetoclax resistance [148]. Finally, a recent clinical study described a similar decreased responsiveness compared to other AML subsets both for FAB-M4 and FAB-M5 patients [164]. Thus, venetoclax resistance is associated with a genetic heterogeneity and probably also different (degrees of) monocytic AML cell differentiation.*Venetoclax resistance can be mediated by various mechanisms.* A recent study combined transcriptomic and ex vivo venetoclax responses to identify individual genes which expression was associated with venetoclax resistance in primary AML cells [165]. They identified four different venetoclax resistance clusters. (i) Their first cluster was associated with monocytic AML cell differentiation; this cluster was also associated with the decreased expression of Bcl-2 together with the increased expression of MCL-1 and BCL-2A1; activation of several metabolic pathways (glycolysis, oxidative phosphorylation and fatty acid metabolism); activation of PI3K-mTOR signaling and susceptibility to the inhibition of cyclin-dependent kinases. In contrast, on the other hand, the three other venetoclax resistance clusters had in common gene enrichment consistent with more immature leukemic cells. (ii) Cluster 2 was associated with FAB-M2 morphology, adverse ELN classification and NRAS mutations; (iii) cluster 3 showed the enrichment of *TP53* mutations and signs of erythroid differentiation, (iv) whereas cluster 4 was associated with favorable ELN classification. Taken together, these observations suggest that venetoclax resistance can be mediated by various mechanisms and that the mechanisms behind FAB-M4/M5-associated resistance differ from other patients. This hypothesis is further supported by another recent study [149] describing that certain resistance mechanisms in primary AML cells reflect a more general resistance to various pharmacological agents independent of the drug target, whereas certain mechanisms for resistance/susceptibility to venetoclax can be independent/different from this general responsiveness.*Erythroid/megakaryocytic differentiation and venetoclax resistance.* An experimental study described that AML cells with erythroid/megakaryocytic differentiation were dependent on antiapoptotic Bcl-xL (encoding the Bcl-2L2 molecule) rather than Bcl-2 or Mcl-1 for survival [156]. These cells were also susceptible to the Bcl-xL-selective inhibitors A-1331852 and navitoclax, whereas they were resistant to the Bcl-2 inhibitor venetoclax. Bcl-xL inhibition also caused extensive killing of such cells in AML xenograft models. Thus, venetoclax resistance due to an altered balance between Bcl-2 family members is not specific for monocytic AML cell differentiation.

Taken together, these studies support the following main conclusions. First, FAB-M4/M5 monocytic AML is associated with venetoclax resistance, but the association seems to be weaker for high-risk/relapsed AML [160,164]. Second, decreased reliance on Bcl-2 for survival and metabolic regulation is one of the mechanisms behind the decreased responsiveness in monocytic AML [164,165,166,167], but other mechanisms, including more general mechanisms such as mechanisms associated with specific genetic abnormalities, also contribute even in monocytic AML [160]. Third, differentiation-associated decreased reliance on Bcl-2 for metabolic regulation and survival is not a specific mechanism for monocytic differentiation. Finally, these mechanisms should also be considered for venetoclax-based combination therapy (see also Section 7) [157].

## 7. AML Cell Differentiation and Targeted Therapies: Effect of Differentiation on the Antileukemic Efficiency and Induction of Differentiation by Targeted Therapies

The importance of AML cell differentiation and differentiation induction for the antileukemic effects of several promising strategies for molecular targeting are summarized in Table 5 and are described more in detail in this section. For many of these strategies, monocytic differentiation is important.

### 7.1. Bromodomain Inhibitors: Increased Antileukemic Effect in Monocytic AML Cells

The Bromodomain and extra-terminal domain (BET) protein family includes four members (i.e., BRD2–4 and BRDT) that function as epigenetic regulators, and BET4 has been identified as a growth-enhancing mediator in AML [168,169]. Experimental studies have demonstrated that monocytic differentiation of the AML cells is associated with sensitivity to BET inhibitors; the monocytic differentiation regulators SPI1 (Spi-1 proto-oncogene), FOS (Fos proto-oncogene, AP-1 transcription factor subunit), JUNB (JunB proto-oncogene, AP-1 transcription factor subunit) and AHR (aryl hydrocarbon receptor) seem to be of particular importance for the responsiveness and show high levels in monocytic AML cells [169]. Thus, in contrast to Bcl-2 inhibitors that are less effective in monocytic leukemia, the BET inhibitors are more effective in this subset. Undifferentiated AML cells show high levels of Bcl-2 and/or CDK2/4/6, and these mediators seem to be involved in the development of BET inhibitor resistance in undifferentiated cells; inhibition of these mediators then resensitizes cells to BET inhibitors. On the other hand, AHR signaling with altered transcriptional regulation seems to be an important characteristic of monocytic AML, and BRD4 then acts as a cofactor for certain hematopoietic transcription factors in these cells [169].

### 7.2. Histone Deacetylase Inhibitors: Epigenetic Modulation and Differentiation Induction

Histone deacetylases (HDACs) are regulators of cancer cell differentiation, and HDAC inhibitors seem to induce AML cell differentiation [59,170,171]. Studies in experimental models have shown that HDAC inhibition decreases AML cell proliferation through cell cycle arrest; decreases expression of the stem cell marker CD117; increases mRNA levels of hematopoietic transcription factors involved in differentiation; increases expression of the myeloid differentiation marker CD11b/CD18 (integrin αMβ2) and causes morphological signs of neutrophil/monocytic differentiation (i.e., reduced nuclear:cytoplasmic ratio, condensation of nuclear chromatin together with infrequent detection of the nucleolus and increased azurophilic granulation) [172]. However, other mechanisms than differentiation induction may also contribute to the antileukemic effects of HDAC inhibitors, including reduced DNA repair [173], altered NF-κB signaling [174] and the modulation of mitochondrial metabolism/pro-survival mechanisms (see Section 5.8).

Several clinical studies have demonstrated that AML therapy including the HDAC inhibitor valproic acid can have clinically relevant antileukemic effects [175,176,177]. A previous clinical study described valproic acid-induced modulation of gene expression profiles of primary AML cells affecting intracellular pathways involved in the regulation of cellular differentiation, as well as the regulation of cell cycle progression, apoptosis and tumor suppression (e.g., the retinoblastoma tumor suppressor and checkpoint signaling in response to DNA damage).

Panobinostate is used in the treatment of multiple myeloma; it has also been tried in the treatment of AML, but as discussed in a recent review, the efficiency can be questioned, and the toxicity seems to be a problem [170].

The HDAC inhibitor I1 has been investigated in the mixed-lineage leukemia (MLL) gene rearranged AML cell lines [178]. The agent had an antiproliferative effect by promoting cell differentiation together with cell cycle block in the G0/G1 phase. These effects were associated with increased acetylation of histone H3 and H4. Similar effects in AML cell lines were also seen for the HDAC inhibitor I3 in AML cells with t(8; 21) translocation and MLL rearranged cell lines [179]; this study also described activation of the VEGF/MAPK (mitogen-activated protein kinase) signaling pathway. Finally, I13 seemed to modulate the antigen processing and presentation pathway [180].

Thus, studies in experimental models and of cells derived from patients receiving antileukemic therapy including HDAC inhibition suggest that differentiation induction is part of the HDAC inhibitor effect on human AML cells.

### 7.3. Lysine Demethylase 1 (LSD1/KDM1A) Inhibitors: Monocytic Differentiation in MLL-AML

The lysine demethylase 1 (KDM1A/LSD1) protein is a component of several histone deacetylase complexes, but it also silences genes and induces differentiation by functioning as a histone demethylase [181]. LSD1 is important for the differentiation block in MLL variants of AML; preclinical studies have shown that LSD1 inhibitors induce monocytic differentiation with morphological signs together with decreased expression of the CD34 stem cell marker and increased expression of CD14, CD36 and CD86 [181,182]. Finally, a recent study described that LSD1/KDM1A inactivation enhances differentiation and promotes the cytotoxic response across AML subtypes when combined with all-trans retinoic acid [183].

Several LSD1 inhibitors have been developed and are in early clinical trials for various diseases [184]. LSD1 inhibition was tested in a Phase I study including 14 patients with relapsed/refractory AML [185]. Monotherapy with the inhibitor ladademstat then had antileukemic activity with reduction of the blast numbers (one complete remission with incomplete normal cell reconstitution); signs of differentiation were observed and the most important toxicities were myelosuppression, infections, mucositis and diarrhea.

### 7.4. Inhibition of DOT1-like Histone Lysine Methyltransferase: Differentiation Induction in MLL-Rearranged and DNMT3A-Mutated AML

Inhibition of the DOT1-like histone H3K79 methyltransferase (DOT1L) is regarded as a possible therapeutic strategy, especially in MLL-rearranged AML [186] but also AML with mutations in DNA methyltransferase 3A (*DNMT3A*) [187,188]. In vitro studies have shown that DOT1L inhibitors induce both morphological (decreased nucleus to cytoplasm ratio, indented nuclei, less basophilic and vacuolated cytoplasm) and molecular (CD11b induction) signs of myeloid differentiation in such AML cells [186,187,188].

DOT1L inhibition as a monotherapy was investigated in a clinical study of patients with advanced hematological malignancies, including relapsed/refractory MLL-rearranged AML [189]; complete remission was seen only for two patients, and the most common adverse events were fatigue (39%), nausea (39%), constipation (35%) and febrile neutropenia (35%). The authors concluded that monotherapy had an acceptable toxicity, but the antileukemic effect was regarded as modest. However, experimental studies suggest that combined DOT1L and MLL-menin inhibition enhances both the differentiation induction and killing of MLL-rearranged AML [188].

### 7.5. Inhibition of the Scaffold Protein Menin: An Effective Strategy in Certain AML Subsets

Menin is a tumor suppressor and a co-factor in leukemogenesis through its interaction with the N-terminal part of MLL that is maintained in different MLL fusion proteins [190]. It also contributes to leukemogenesis in *NPM1-Ins* AML through binding to specific chromatin targets that can be occupied both by NPM1 and MLL. Thus, menin inhibition is probably an effective antileukemic strategy but only in certain AML subsets:*MLL-rearranged AML.* Inhibition of the menin/MLL interaction is a possible strategy in MLL-rearranged AML. VTP50469 is a potent, highly selective and orally bioavailable inhibitor that displaces menin from protein complexes and inhibits its chromatin binding [191]. This loss of MLL binding leads to changes in gene expression, cellular differentiation and regulation of apoptosis, and these antileukemic effects have been observed even in AML xenograft models. Furthermore, the knockout/degradation of menin or treatment with the menin inhibitor SNDX-50469 reduce MLL fusion protein-induced AML cell differentiation and reduces AML cell viability, as well as the Bcl-2 and CDK6 levels, but despite the reduced Bcl-2 levels, the combination of SNDX-50469 with venetoclax or the CDK6 inhibitor abemaciclib has synergistic antileukemic effects for patient-derived AML cells harboring *MLL1* rearrangements or *NPM1-Ins* [192].*FLT3-ITD AML.* Pharmacological inhibition of the menin-MLL complex caused specific changes in gene expression with downregulation of both the MEIS1 transcription factor and its target FLT3 [193]. Combined menin-MLL and FLT3 inhibition had synergistic antiproliferative and proapoptotic effects in models of human *NPM1*-mutated or *MLL*-rearranged AML with additional *FLT3-ITD* [194]. Importantly, AML cells from patients with both *NPM1* and *FLT3* mutations showed significantly better responses to combined menin/FLT3 inhibition than to single-drug treatment.*NUP98 translocations.* The menin-MLL1 interaction is essential in AML with translocations involving the *NUP98* (nucleoporin 98 and 96 precursor) gene, and studies of NUP98 fusion leukemia in animal models have shown that inhibition of the menin-MLL1 interaction by VTP50469 has an antileukemic effect [195]. This inhibition upregulates various differentiation markers such as CD11b by removing MLL1 and NUP98 fusion proteins from chromatin sites at genes that are essential for the malignant phenotype. The antileukemic and differentiation-inducing effects were also observed in patient-derived xenografts of *NUP98* fusion-driven AML.*UBTF abnormalities.* AML with *UB*TF (upstream binding transcription factor) tandem duplications (UBTF-TDs) has a transcriptional signature with similarities to *NUP98*-rearranged and *NPM1*-mutated AML; the primary cells from *UBTF-TD* AML are sensitive to the menin inhibitor SNDX-5613 that has antiproliferative and differentiation-inducing effects in these cells [196].*MN1 translocations.* Translocations involving meningioma-1 (*MN1*) occur in an AML subset and result in a high expression of either the full-length MN1 protein or a fusion protein including most of the N-terminus of MN1 [197]. Menin is essential for the self-renewal of MN1-driven AML, and pharmacological inhibition of the MLL-menin interaction also has antiproliferative and differentiation-inducing effects in this AML variant when tested in experimental in vitro and xenograft models.

Early clinical studies suggest that the overall response rate to this therapy for patients with MLL or NPM1 abnormalities is 40–50%, and the complete remission rate is 20–35%; this is true even for heavily pretreated *NPM1* mutated patients [190]. Differentiation syndrome has been an important adverse event. Thus, menin targeting is a promising therapeutic strategy in certain AML subsets, and there seems to be an increased efficiency when menin inhibition is combined with venetoclax or Flt3 inhibitors.

### 7.6. Inhibitors of the Nuclear Exporter Exportin-1 (XPO1): Effects on NPM-Ins, as well as Other Leukemogenic Proteins

The cytoplasmic dislocation of the *NPM1*-mutated protein is critical for its oncogenicity, and reduced cytoplasmic localization is associated with an antiproliferative effect, G1 arrest and induction of differentiation [198]. The differentiation can be detected as morphological signs of monocytic or granulocytic differentiation together with the increased expression of molecular markers (CD11b and CD14). The differentiation induction has been detected in both by in vitro studies and patient-derived xenograft models.

Inhibition of nuclear export by the XPO1 inhibitor selinexor is associated with differentiation induction and prolonged survival in various murine AML models; this agent inhibits the nuclear export of not only mutated *NPM1* but also several other molecules involved in leukemogenesis [198]. It will also upregulate the purinergic receptor P2Ry2 in AML cells and thereby activate PI3K-AKT signaling; inhibition of this pathway has therefore potentiated the antileukemic effects of selinexor in experimental models [199]. Although selinexor induces differentiation, it can have synergistic effects when combined with venetoclax, at least in certain models/patients; venetoclax then increases selinexor-induced DNA damage, probably through DNA repair inhibition [200].

Selinexor was investigated in early clinical AML studies. First, selinexor monotherapy can have an antileukemic effect in relapsed/refractory AML, and this effect is probably due to its general effect on the nuclear–cytoplasmic transport of various proteins involved in leukemogenesis [201]. Second, the drug seems to have acceptable toxicity when combined with intensive conventional cytotoxic therapy (mitoxantrone, etoposide and cytarabine) in relapsed/refractory AML; the overall response rate for 23 patients was 43%, and the most common grade ≥3 non-hematologic toxicities were febrile neutropenia, catheter-related infections, diarrhea, hyponatremia and sepsis [202]. Third, experimental studies suggest that hypomethylating agents can increase the antileukemic effect of selinexor [203]. Finally, experimental studies of second-generation exportin inhibitors seem to be very effective against AML but are possibly less toxic [204].

### 7.7. Ribosome Targeting Seems to Overcome Venetoclax Resistance

A recent study described that the ribosome targeting antibiotic tedizolid can overcome venetoclax resistance [205]. This combination was then tested both in vitro and in patient xenografts models. Superior antileukemic activity was observed by the combination compared to single-agent treatment, but the triple combination of venetoclax, azacitidine and tedizolid was even more effective than the dual combination. It is not known whether ribosome targets monocytic-associated or other resistance mechanisms.

### 7.8. Proteasome Inhibition: Is the Antileukemic Effect Associated with Monocytic Differentiation?

A previous experimental study described an association between the expression of monocytic differentiation markers by primary AML cells and a high sensitivity to bortezomib-mediated apoptosis [206]. However, this association was not observed in another experimental study testing the effects of bortezomib and another proteasome inhibitor on primary AML cells [207], whereas a study of AML cell lines described that bortezomib sensitized myelomonocytic AML cell lines to TRAIL-induced apoptosis [208].

### 7.9. Flt3 Inhibition: Differentiation Induction versus Cytotoxicity

Monocytic AML cell differentiation can occur during treatment with Flt3 inhibitors, and this has been described both for gilteritinib and midostaurin [209,210]. Other studies have suggested that granulocytic [211], and even erythroid [194], differentiation may also occur. Clinical differentiation syndrome can develop during treatment with Flt3 inhibitors [168], but this is observed only for approximately 3% of patients [212]. It has therefore been suggested that there are two types of responses to FLT3 inhibition; one type is the induction of differentiation, and the other type is a cytotoxic effect without differentiation but characterized by a reduction of *FLT3*-mutated cells [210,211]. A previous study suggested that CDC25A downregulation induced by FLT3 inhibition is involved in the monocytic differentiation response that is characterized by an increased expression of several monocytic markers (e.g., CD11b and CD14) [213]. However, this mechanism cannot explain the induction of granulocytic or erythroid differentiation.

### 7.10. The TRAIL Agonist Eftozanermin: Combined Treatment with Venetoclax

Eftozanermin is a TRAIL agonist that has been combined with venetoclax to achieve stimulation of both the extrinsic and intrinsic pathways of apoptosis. A recent study reported experimental and clinical effects of this combination [214]; the experimental investigations included both ex vivo studies and patient-derived xenograft models showing increased antileukemic effects of the combination. The drug combination was also examined in relapsed/refractory AML in a Phase I study; eftozanermin monotherapy seemed to have a limited antileukemic effect, but the overall response rate was 30% for 23 patients treated with venetoclax plus eftozanermin. The responses included four patients with complete remissions, and the median duration of the responses was 7.8 months. Treatment-related adverse events ≥3 were seen for 30% of patients, increased serum levels of liver enzymes being most common. These results suggest that venetoclax combined with eftozanermin can have an antileukemic effect even in relapsed/refractory AML, and the toxicity of the combined treatment seems to be acceptable.

### 7.11. Aryl Hydrocarbon Receptor Agonists and Induction of Monocytic/Granulocytic Differentiation

The aryl hydrocarbon receptor (AHR) functions as a ligand-activated transcription factor (for references, see [215]). AHR agonists can inhibit AML stem cells and induce leukemic stem cell differentiation by the induction of monocyte and/or granulocyte markers in patients with Flt3-ITD, and the agonists seem to have synergistic effects with the Flt3 inhibitor gilteritinib [215]. The ligand-induced differentiation involved the AML stem/progenitor cells and was characterized by increased levels of CD11b, CD14 and CD15, as well as morphological signs of differentiation (i.e., altered nuclear morphology) and decreased AML cell proliferation. However, it should be emphasized that these effects seemed to vary between patients.

### 7.12. IDH Mutations and IDH Inhibition: Venetoclax Sensitization and Differentiation Induction

*IDH1* and *IDH2* mutations are associated with the differentiated FAB-M2 phenotype [104,216]. Differentiation syndrome is seen in 15–20% of patients during IDH inhibitory treatment; the risk factors for its development are high leukemia cell burden and concomitant *TET2* and *SRS2* mutations, and the syndrome is characterized by leukocytosis with increased neutrophil and monocyte levels, but the syndrome is not associated with improved outcome [217,218,219]. Furthermore, enasidenib is an *IDH2* mutant inhibitor that can increase the sensitivity to venetoclax; this sensitizing effect was observed both in vitro and in human AML xenograft models and seemed to depend on whether the leukemic cells differentiated (i.e., decreased c-kit/CD117 and increased granulocytic marker GR-1) in response to the enasidenib treatment [220]. Thus, there seems to be combined monocytic and neutrophil differentiation. It is difficult to judge whether the differentiation represents a further combined myelomonocytic differentiation of individual cells (i.e., similar to pretreatment FAB-M4) or whether monocytic and further neutrophil differentiation (similar to pretreatment of the neutrophil FAB-M2 versus the monocytic, respectively) occurs in separate AML cell subclones.

### 7.13. Inhibition of Pyrimidine Biosynthesis and Purine Metabolism Induces Differentiation

Pyrimidine bases are parts of many cellular macromolecules and not only DNA and RNA, and dihydroorotate dehydrogenase (*DHODH*) is regarded as a rate-limiting enzyme for pyrimidine nucleotide synthesis [221]. Purine/pyrimidine are synthesized from amino acids, and AML stem cells show increased uptake, steady-state levels and the catabolism of amino acids and are also reliant on amino acid metabolism for oxidative phosphorylation and survival [222]. The inhibition of DHODH can induce differentiation of the AML cell lines and primary AML cells; the signs of differentiation include an increased expression of CD11b, CD13, CD14 and CD33 (including CD13^+^CD33^+^ double-positive cells); morphological signs of differentiation with a lower nucleocytoplasmic ratio, condensed chromatin and increased nuclear lobulation and functional signs, as well as gene expression profiles, consistent with neutrophil differentiation [223]. These signs are associated with the induction of apoptosis. The differentiation/proapoptotic effect is seen across genetic AML subtypes [223,224] but varies between patients [223]. Antileukemic effects have been described even for patients with relapsed/refractory AML [223], although resistance then seems more common [216], and this is possibly due to a high expression of CDK5 [224] and increased fatty acid metabolism [222]. The effect of such enzyme inhibition on normal hematopoietic cells is weak, and murine studies, as well as the initial clinical studies, suggest that the safety profile of these inhibitors is acceptable [223].

A recent study described that monocytic AML stem cells are characterized by a different regulation of one carbon metabolism/purine synthesis; these experimental studies suggest that inhibition of this metabolic pathway (i.e., cladribine therapy) may be used for a more specific targeting of monocytic AML stem cells [158].

### 7.14. Therapeutic Targeting of Energy Metabolism by Direct Inhibition of Metabolic Regulators or Inhibition of the Metabolic Sensor/Regulator PI3K-Akt-mTOR Pathway

Targeting of the mitochondrial metabolism with the reprogramming of glycolysis, glutaminolysis, oxidative phosphorylation or ATP production is regarded as possible strategies for cancer treatment [78,92,225], including AML therapy [86]. This strategy may be the most effective in cancer types that show increased oxidative phosphorylation [92]. Metabolic targeting may be effective even against the metabolically fittest cells within heterogeneous cancer cell populations, i.e., ATP high stem cell-like cells that show multidrug resistance and an increased capacity for migration, invasion and metastasis [226].

The following examples support the suggestion that metabolic targeting should also be further investigated in AML. First, CPI-613 is an inhibitor of pyruvate dehydrogenase and α-ketoglutarate dehydrogenase and can sensitize AML cells to conventional cytotoxic drugs [227]. This effect has been observed in cell lines, as well as animal models, and combined treatment with cytotoxic drugs has also been investigated in an early Phase I study [227]. Second, metabolic targeting can induce AML cell differentiation. Pharmacological inhibition of the mitochondrial electron transport chain complex III by antimycin A inhibits proliferation and induces AML cell differentiation, probably through the inhibition of dihydroorotate dehydrogenase and pyrimidine biosynthesis (see Section 7.13) [228]. Combining complex III inhibition with the dihydroorotate dehydrogenase inhibitor brequinar or the pyrimidine salvage inhibitor dipyridamole also had synergistic effects [228]. Third, AML stem cells seem to be more susceptible to reactive oxygen species than normal hematopoietic stem cells, and the induction of oxidative stress through agents that modulate microenvironmental levels of reactive oxygen species has therefore been suggested as a possible antileukemic strategy [229]. Finally, experimental studies have demonstrated that metformin has an antiproliferative effect in MLL/AF9 AML cells through the inhibition of mitochondrial respiration [230]. In our opinion, it is therefore justified to conclude that metabolic targeting should be further investigated as a possible therapeutic strategy in human AML, and this is also supported by the clinical experience with venetoclax as a metabolic modulator (see Section 6).

An alternative strategy for metabolic modulation may be the use of asparaginase that reduces the systemic glutathione levels and thereby also reduces MCL-1 protein expression (Table 5) [162]; combined treatment of venetoclax and asp arginase was suggested as a possible therapeutic strategy by these investigators. Future studies have to clarify whether the combination of venetoclax with asparaginase to reduce MCL-1 levels or the combination with a MCL-1 inhibitor [167] is more effective. 

The PI3K-Akt-mTOR pathway is regarded as a metabolic sensor/regulator, and monocytic AML cell differentiation is associated with the high activity of this signaling pathway [165]. A recent experimental study suggested that co-targeting of Bcl-2 and PI3K induces Bax-dependent mitochondrial apoptosis in AML (Table 5) [161].

### 7.15. The Possible Therapeutic Targeting of Monocytic Markers

The molecular markers that can be associated with monocytic FAB-M4/M5 differentiation are presented in Table 1 and Appendix A and in Figure 1. Although the experience with targeting of these markers is limited, experimental studies suggest that this may be a possible strategy in the future treatment of human AML. The possible strategies include: *Targeting of Mcl-1.* Monocytic AML cells rely on Mcl-1 rather than Bcl-2 for survival and metabolic regulation, and experimental studies suggest that Mcl-1 targeting is a possible strategy either as a monotherapy or in combination with conventional chemotherapy or Bcl-2 targeting [115,231,232,233,234,235,236]. Mcl-1 inhibition can be achieved either by direct inhibition [115,231,232,233,234] or indirectly through transcriptional or translational suppression [235,236]. However, even though this strategy seems to have an anticancer effect, the risk of cardiotoxicity may make it unfeasible [237].*Integrin targeting.* Blocking of integrin cell surface molecules by monoclonal antibodies is now used as a strategy for the inhibition of immunocompetent cell extravasation/chemotaxis [238]; the blocking of leukemia-supporting AML/endothelial cell interactions in their common bone marrow environment may similarly be used as an antileukemic strategy, either by using integrin-directed antibodies [238] or possibly soluble integrin molecules that compete for the binding on the integrin ligands [239,240]. The inhibition of β2 integrins by small molecular inhibitors may also be possible [241], although this strategy will not be specific for CD11b/CD18 and CD11c/CD18. Furthermore, one would expect the targeting of CD11b/CD18 and CD11c/CD18 to be associated with a risk of infection due to the inhibition of normal monocytes [242], and this may become particularly important for patients receiving AML therapy with hematological toxicity [3]. Finally, the effects of integrins on downstream intracellular signaling are very complex, and many integrins modulate downstream pathways that are also shared with other cell surface molecules/receptors [238].*Targeting of intracellular signaling.* The monocyte markers include several receptor molecules; these markers may be targeted by blocking of their downstream signaling by small molecule inhibitors. This strategy may be possible for PI3K-Akt-mTOR (CD7, CD56 and CD117) [243,244]; NF-κB (CD14) [245]; CD117/c-kit [246]; TIMP1 (CD63) [247,248] and FAK (CD63, CD117 and ITGB1) [249] signaling. It should be emphasized that these pathways are common for various upstream receptors; this approach will thus also block signaling not only from the corresponding monocyte marker/receptor but also other receptors/mediators upstream from the target.*Targeting of cellular function.* Several monocyte markers seem to affect intracellular transport/trafficking, and the targeting of these processes may also be possible [250,251,252]. However, this strategy should also be regarded as nonspecific and will not only affect signaling from/functions of specific monocytic markers.*Immunotherapy.* To the best of our knowledge, there is no firm evidence for an association between the effect of immunotherapy and monocytic AML differentiation.

The identification of possible targets associated with monocytic differentiation should, in our opinion, be based on protein analyses; gene expression studies are less reliable for the identification of possible targets because of the possibility of posttranscriptional regulation and the variations between patients [20,21,22,23,24].

## 8. Coagulopathy and Risk of Thrombosis in AML: A Risk Associated with Differentiation or Differentiation Induction?

There seems to be an increased risk of thrombosis for patients with AML; a recent study estimated the overall risk to be 8.7% over a median follow-up of 478 days and with an even higher risk of 10.4% for elderly patients [253]. Most thrombotic events (66%) occurred before the start of the second chemotherapy cycle, and arterial and venous thromboses had comparable frequencies. Another large study investigated only venous thromboembolism and described a similar frequency [254], whereas a lower frequency was described in a smaller study [255]. Several molecular and cellular mechanisms are involved in the pathogenesis of AML-associated coagulopathy, including AML cell expression of procoagulant proteins, exposure of procoagulant lipids, release of inflammatory cytokines (e.g., IL1β and TNFα) and microparticles and the adhesion to vascular cells, leading to activation of the coagulation cascade, platelets and endothelial cells [256].

A recent study showed a statistically significant association between the high frequency of coagulopathy (i.e., laboratory signs consistent with disseminated intravascular coagulation) and monocytic AML cell differentiation [253]. A high frequency of coagulopathy (26%) for patients with acute monoblastic leukemia was also described in a previous study, and coagulopathy was observed especially for patients with high peripheral blood blast counts [257]. The number of thrombotic events in these studies was too few to conclude whether the frequency of thrombotic events was also higher in monocytic AML.

Monocytic differentiation is induced by various targeted therapies. Future clinical studies should therefore address whether such differentiation induction is associated with coagulopathy/thrombosis, especially for those patients where the treatment induces disease stabilization without remission because most thrombotic events seem to occur before complete remission is achieved for patients receiving intensive therapy [253].

## 9. Discussion

In this review, we discuss the differentiation block as a fundamental characteristic of the leukemic cells in non-APL variants of AML. However, for certain patients, limited signs of differentiation can be seen, for example, morphological and molecular signs of monocytic differentiation [1,2]. However, even for these patients, there is a differentiation block, and the signs of differentiation are limited.

In contrast to patients with APL that show one dominating and a limited number of other genetic abnormalities [258,259], patients with monocytic AML variants are heterogeneous with regards to cytogenetic and molecular genetic abnormalities. However, patients showing monocytic differentiation have other important biological characteristics in common, especially molecular mechanisms for the mitochondrial regulation of energy metabolism and survival/apoptosis. The monocytic FAB-M4/M5 patients can also be divided into various subsets based on biological functions/proteomic profiles. Furthermore, morphological signs of differentiation seem to have a very limited prognostic impact in patients receiving intensive conventional cytotoxic therapy possibly combined with allogeneic stem cell transplantation [11,15,58,59,60]. This is probably due to the strong and general prognostic impact of various genetic abnormalities for patients receiving this type of antileukemic treatment. The additional prognostic impact of the differentiation status may become more important when using the therapeutic targeting of specific molecules when these targets are regulated by molecular networks closely linked to the differentiated AML cell phenotype.

Signs of differentiation, including both monocytic and probably erythroid, as well as megakaryocytic differentiation, can be associated with a resistance to Bcl-2 inhibition. This seems to be due to differentiation-dependent variations in the levels and thereby the relative importance of various antiapoptotic Bcl-2 family member (see Section 6). Differentiation induction, and especially monocytic differentiation, seems to be a part of the therapeutic response to several new targeted therapies that are considered for AML therapy (see Section 7), and for the IDH inhibitor enasidenib, such differentiation induction seems to be necessary for an enasidenib-associated increase in venetoclax sensitivity [220]. Thus, several observations suggest that the induction of AML (stem) cell differentiation is part of the antileukemic effect for many new targeted therapies, and the pretreatment differentiation status may then influence the responsiveness to these therapies. The differentiation induction may influence the effects and possible synergism when using combined targeted therapy as well.

As described in Section 7, several targeted therapies seem to induce AML cell differentiation as part of their antileukemic effect. However, it should be emphasized that the differentiation in these pharmacological studies is generally characterized only by morphology together with the expression of one or only a few of the monocytic differentiation markers (Table 1). A better characterization of the differentiation induction at the molecular and functional levels is therefore needed for these targeted therapies, and for this reason, it is not known to which degree the monocytic differentiation induced by targeted therapies show (all) the same biological characteristics as the pretreatment monocytic differentiation detected by the FAM-M4/M5 classification.

A final question is whether the degree of monocytic AML cell differentiation is decisive for the biological and/clinical impacts of the differentiation. The results from a recent report on responsiveness to venetoclax-based therapy suggest that the prognostic impact of myelomonocytic or monocytic differentiation (i.e., FAB-M4 versus FAB-M5) is similar, at least for this therapeutic strategy [164]. This observation suggests that these two AML subsets share common mechanisms with regards to the mitochondrial regulation of apoptosis/survival and cell metabolism. Future studies have to clarify whether the induction of monocytic differentiation by targeted therapy will be associated with increased frequencies of AML complications, e.g., the development of coagulopathy.

## 10. Conclusions

Signs of differentiation do not have any important prognostic impact for patients with non-APL variants of AML receiving conventional intensive cytotoxic therapy eventually combined with allogeneic stem cell transplantation. In contrast, signs of differentiation (including monocytic differentiation) are associated with responsiveness to several new therapeutic strategies based on the targeting of specific molecules or molecular networks/pathways, and the induction of monocytic differentiation may also be part of the antileukemic effects for such therapeutic strategies.

## Figures and Tables

**Figure 1 ijms-25-06356-f001:**
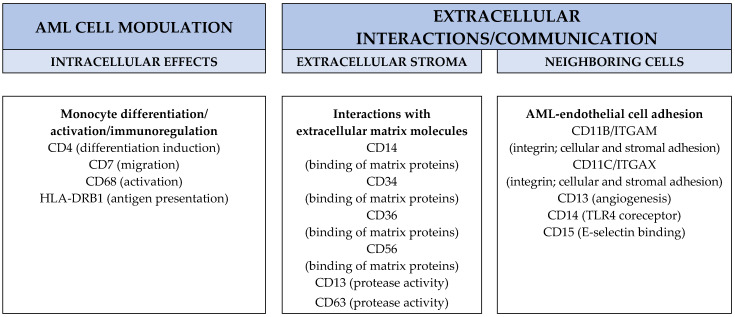
Functional classification and characterization of monocytic molecular markers expressed by FAB-M4/M5 AML cells and based on important molecular functions. Keywords describing selected important ligands and/or functions (based on the Gene database Home—Gene—NCBI (https://www.nih.gov/), accessed on 13 April 2024) are given in parentheses.

**Table 1 ijms-25-06356-t001:** Characteristics of AML patients with monocytic differentiation (FAB-M4/M5) of their leukemic cells according to the WHO 2016 classification system [4]. The table presents these criteria for monocytic AML blast differentiation (i.e., corresponding to the FAB-M4/M5 classification) used to define the subsets of acute myelomonocytic leukemia (FAB-M4) and acute monoblastic and monocytic leukemia (FAB-M5) among AML not otherwise specified. The main criterion is leukemia cell morphology, whereas histochemistry and immunophenotyping are used as supportive diagnostic tools.

Characteristic	FAB-M4; Acute Myelomonocytic Leukemia	FAB-M5; Acute Monoblastic and Monocytic Leukemia
<Morphology of the AML cells	(i) Both monocytes and monocyte precursors, as well as neutrophils and their precursors, constitute ≥20% of bone marrow nucleated cells (ii) A high percentage of monocytic cells may also be present in peripheral blood	(i) Monoblasts, promonocytes and monocytes constitute ≥80% of bone marrow nucleated cells, but a minor neutrophil component ≤20% may be present (ii) Acute monoblastic leukemia contains ≥80% of monoblasts among the monocytic cells (iii) Acute monocytic leukemia contains mainly promonocytes and monocytes among the monocytic cells
Histochemistry (the AML cells)	(i) Leukemic cells are positive for myeloperoxidase (≥3% of the blasts) (ii) Leukemic cells are usually positive also for non-specific esterase and naphtol chloroesterase	Monocytes and promyelocytes are usually positive only for non-specific esterase but not myeloperoxidase
Immunophenotype of the AML cells	(i) Several blast populations are often identified, and they show variable positivity for the myeloid markers CD13, CD15, CD33 and CD65 (ii) Most cases also express HLA-DR, and a minority of cases express CD7 (iii) At least one blast population being positive for monocytic markers, e.g., CD4, CD11b, CD11c, CD14, CD36, CD63, CD64, CD68 and/or lysozyme. (iv) Monocytic differentiation may also be characterized by co-expression of CD15 or CD36 or strong CD64 expression (v) A subpopulation of immature CD34^+^/CD117^+^ blasts is often detected	(i) Usually expression of the myeloid markers CD13, CD15, CD33 and CD65 (ii) Most cases express HLA-DR, and a minority express CD7 or CD56 (iii) There is usually expression of at least two monocytic markers, e.g., CD4, CD11b, CD11c, CD14, CD36, CD64 or lysozyme (iv) CD34 is positive for a minority, CD117 is more often expressed and CD68 and CD163 are often expressed

**Table 2 ijms-25-06356-t002:** A summary of molecular markers not being used as markers of monocytic FAB-M4/M5 AML according to the WHO 2016 classification [4] but showing different expression in monocytic FAB-M4/M5 AML cells compared to other AML subsets. Increased/decreased expression in monocytic AML is indicated by arrows. Question marks indicate that the available results are conflicting, even though some studies have described the indicated difference; the arrows indicate whether levels are increased (↑) or decreased (↓).

**Protein levels of T-cell costimulatory and checkpoint molecules**
↑ CD86?
↑ PD-L1?
**Cell surface protein levels of cytokine receptors**
↑ GM-CSF receptor
↑ Flt3
↓ SCF receptor/CD117
**Release of soluble protein mediators**
↑ Several CCL and CXCL chemokines
↑ Several Interleukins
↑ Several proteases and protease regulators
↑ Several hematopoietic growth factors

**Table 3 ijms-25-06356-t003:** A summary of the molecular crosstalk between regulators of cellular metabolism and cell survival (for references and more detailed discussion, see the text (Section 5)).

Mediator	Effect of the Mediator on Cell Metabolism	Effect of Cell Metabolism on the Mediator
Bax	(i) Regulates levels of reactive oxygen species (ii) Modulates ceramide/lipid metabolism (iii) Regulates mitochondrial remodeling together with Bak	
Cytochrome C	Passing electrons from complex III to complex IV in the electron transport chain	Glucose-stimulated glutathione production inactivates, but reactive oxygen species activate cytochrome c
Bcl-2	(i) Increases mitochondrial respiration (ii) Regulates cytochrome c activity and the balance between its Va and Vb subunits	Metabolic stress modulates Bcl-2 activity
Mcl-1	(i) Promotes ATP production (ii) Regulates AML cell metabolisms, including the tricarboxylic acid cycle, glycolysis and pentose phosphate pathway	Targeted for degradation in the absence of glucose
Bcl-xL	(i) Reduces citrate and thereby acetyl-CoA levels and protein acetylation (see also Sirtuins below) (ii) Increases ATP generation through increased oxidative phosphorylation (iii) Lowers mitochondrial oxygen consumption and modulates tricarboxylic acid cycle activity	
Bad	Regulates glucose metabolism	
Puma		Its expression is induced/increased by glucose deprivation
Noxa	(i) Promotes glycolysis (ii) Regulates the balance between glycolysis and the pentose phosphate pathway	The levels are decreased by glucose
TP53	(i) Suppresses glycolytic flux and upregulates the tricarboxylic acid cycle (ii) Modulates the balance between the glycolytic and the pentose phosphate pathways (iii) Downregulates lipid synthesis (iv) Increases oxidative phosphorylation (v) Maintains mitochondrial integrity (vi) Reduces deleterious effects of oxidative stress	There is a crosstalk between the metabolic sensor mTORC1 and TP53
Sirtuins	These lysine deacetylases modulate cellular metabolism, e.g., lipid metabolism and urea cycle regulation	Protein acetylation is also influenced by availability of acetyl-CoA from the tricarboxylic acid cycle

**Table 4 ijms-25-06356-t004:** Venetoclax resistance in human AML: A summary of important available data.

**Clinical studies of venetoclax resistance in AML**	
Morphological differentiation	Monocytic differentiation is associated with venetoclax resistance [7,8,88,151,159]. Patients with newly diagnosed monocytic AML show significantly lower remission rate and overall survival compared to non-monocytic subsets and with no significant differences between the FAB-M4 and FAB-M5 subtypes [164]. Lower remission rate for monocytic variants was also observed for relapse/refractory AML patients, but the overall survival did not differ for these patients [164].
Gene expression profiles	Gene expression profiling of primary AML cells detected four patient clusters associated with venetoclax resistance. (i) One cluster showed an enrichment of monocytic AML cells, as well as transcriptional activation of PI3K-AKT-mTOR signaling and energy metabolism pathways; these cells showed sensitivity to mTOR and CDK inhibition. (ii) The second cluster was enriched for *NRAS* mutations and associated with decreased *HOX* expression. (iii) The third was characterized by *TP53* mutations and the expression of erythroid markers, and (iv) the last cluster showed overexpression of interferon signaling and high rates of *DNMT3A* mutations. A patient’s AML cell population may consist of cell subsets mimicking different resistance clusters [165].
Bcl-2 family	The combinatorial ratio/score of the antiapoptotic mediators Bcl-2, Bcl-xL and MCL-1 expression in leukemic stem cells can predict the clinical responses to venetoclax [160].
Senescence	Regulation of senescence together with PD1 signaling and NADP oxidase activity seems to be involved in the clinical venetoclax resistance associated with monocytic differentiation [166].
AML stem cells	There seems to be a selective outgrowth of monocytic AML cells at the time of progression during venetoclax therapy [7]. This progression arises from leukemia stem cells that seem to differ from other primitive leukemic stem cells with regards to their immunophenotype (CD34^−^, CD4^+^, CD11b^−^, CD14^−^ and CD36^−^); their reliance on purine metabolism and a different chemosensitivity profile. These monocytic stem cells may co-reside with other subsets of more primitive leukemic stem cells [158].
**Other factors than monocytic AML cell differentiation associated with venetoclax susceptibility/resistance**	
Mutations	Sensitivity to venetoclax seems to be associated with *NPM1*, *IDH1/2*, *TET2*, *RUNX1* and *ASXL1* mutations, whereas venetoclax resistance is associated with *FLT3-ITD*, *TP53*, *K/NRAS* or *PTPN11* mutations [139,140,148,159].
Karyotype	Adverse karyotype according to the ELN classification is also associated with resistance [141].
Clonal heterogeneity	Clonal heterogeneity is associated with venetoclax resistance [143,144,164].
**Experimental studies**	
Differentiation	There is an association between venetoclax resistance and monocytic differentiation [163].
Bcl-2 family	MCL-1 inhibition is an effective antileukemic strategy in venetoclax-resistant AML cells and patient-derived xenografts, and molecular profiling can be used to predict the efficiency of Mcl-1 versus Bcl-2 inhibition [167]. Resistance to venetoclax is associated with a low Bcl-2/Mcl-1 ratio [7]. Myelomonocytic, as well as upregulated, Bcl-2A1 and CLEC7A confer resistance [148].
Mutations	Mutations of *PTPN11* and *KRAS* also confer resistance to venetoclax-based therapy [148].
Ribosome	The ribosomal protein S6 kinase alpha-1 mediates venetoclax resistance in monocytic AML [150].
Metabolism	Metabolic effects of venetoclax involving oxidative phosphorylation seem to be important for its antileukemic effects in Bcl-2-dependent AML cells [87,88].
**Combination of venetoclax with other targeted therapies**	
PI3K-mTOR	Combination of venetoclax with PI3K-mTOR inhibition has synergistic effects, and this increases sensitivity depends especially on BAX but also on BAK expression [161].
Asparaginase	Combination of venetoclax and pegylated asparaginase has synergistic effects; this is possibly due to reduction of glutathione levels and thereby also reduced Mcl-1 expression [162].
Mcl-1	Combination of Bcl-2 and Mcl-1 inhibition has synergistic effects [167].

**Table 5 ijms-25-06356-t005:** Therapeutic targeting of specific molecular mechanisms in human AML, and overview of the differentiation-inducing effects of various targeted therapies (for detailed description and references, see the main text in Section 7).

Target	Effect of AML Cell Differentiation on Targeted Therapy or Effects of Targeted Therapy on AML Cell Differentiation
*Epigenetic targeting/histone modification*	
BET	Less effective in immature/undifferentiated AML cells, and monocytic markers correlates with sensitivity. The inhibitors induce monocytic differentiation.
Histone deacetylase	HDAC inhibition induces myeloid differentiation with induction of the transcription factors associated with differentiation and reduced expression of CD117, together with increased expression of CD11b/CD18.
Lysine demethylase 1	LSD1 is important for the differentiation block in MLL variants of AML; LSD1 inhibitors induce morphological signs of monocytic differentiation; decreased CD34 expression and increased expression of CD14, CD36 and CD86.
DOT1L	In vitro studies have shown that DOT1L inhibitors induce both morphological and molecular signs of myeloid differentiation in *MLL*-rearranged and *DNMT3A*-mutated AML.
Menin	Menin inhibition is a possible strategy that modulates AML cell differentiation, especially in *MLL*-rearranged and *NPM1*-mutated AML but possibly also AML variants with *FLT3-ITD*, *NUP98* translocations, *UBTF* tandem duplications and *MN1* translocations.
*Molecular transport*	
Exportin 1	Exportin 1 inhibition in patients with *NPM1* mutation induces morphological signs of monocytic or granulocytic differentiation together with increased CD11b and CD14 expression.
*Protein synthesis and modification*	
Ribosome targeting	Combination of venetoclax (decreased susceptibility in FAB-M4/M5) with the ribosome-targeting tedizolid increases the antileukemic effects compared to monotherapy.
Proteasome	Increased antileukemic effect in monocytic AML cells.
*Intracellular signaling*	
FLT3	Monocytic differentiation seems most common; even granulocytic and erythroid differentiation are described.
TRAIL agonist	Combination of venetoclax (decreased susceptibility in FAB-M4/M5) with the TRAIL agonist eftozanermin increases the antileukemic effects compared to monotherapy.
Aryl hydrocarbon receptor	Receptor agonists can inhibit AML stem cells and induce AML stem cell differentiation by the induction of monocyte and/or granulocyte markers in patients with Flt3-ITD characterized by increased levels of CD11b, CD14 and CD15, as well as morphological signs of differentiation.
*Cellular metabolism*	
IDH1/IDH2	IDH mutations are associated with the FAB-M2 phenotype. Induction of monocytic and or granulocytic differentiation.
Pyrimidine biosynthesis	Inhibition of the rate-limiting step induces morphological AML cell differentiation with increased expression of CD11b, CD13, CD14, and CD33 and functional signs, as well as gene expression profile modulation consistent with neutrophil differentiation.
Energy metabolism	Inhibition of the mitochondrial electron transport chain complex III A inhibits proliferation and induces AML cell differentiation. The antileukemic effects of targeting antiapoptotic and metabolism-regulatory Bcl-2 family members depend on the differentiation status of the AML cells.

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
