# Peer review of "Monocytic Differentiation in Acute Myeloid Leukemia Cells: Diagnostic Criteria, Biological Heterogeneity, Mitochondrial Metabolism, Resistance to and Induction by Targeted Therapies"

_ijms, 2024, doi:10.3390/ijms25126356_

Round 1

Reviewer 1 Report

Comments and Suggestions for Authors

The review by Bruserud et al provides a detailed description of monocytic differentiation in acute myeloid leukemia (AML), encompassing morphology, molecular markers, cytogenetic and genetic profiles, response to current therapy of M4-M5 AML and the role of monocytic differentiation in response to targeted therapies in AML.

The review provides a very interesting point of view in the contextualization of the monocytic differentiation in acute myeloid leukemia (AML). However, the focus should be kept throughput the manuscript.

Please add percentages of the occurrence of the diverse mutations/cytogenetic abnormalities that can be found in M4 and M5 AML.

The paragraph on “Expression of costimulatory molecules and checkpoint ligands” has a partial focus on M4-M5 AML. This should be revised to be focused and discuss potential therapeutic strategies related to immune checkpoint inhibition.

In addition to surface molecules, the transcriptional program/signatures that characterize M4-M5 AML should be discussed. This topic should also drive the discussion of experimental therapies (preclinical drug studies) that have tested targeted therapies against those markers in AML (please add an additional paragraph).

One issue in M4-M5 AML is the distinction between normal and leukemic monocytic cells. This topic, that is currently mentioned in paragraph 2.7, should be addressed in details in paragraph 2.6 (e.g. surface markers, cytokine release…). Moreover, a figure showing how the molecules expressed by M4-M5 AML are involved in the interaction with the microenvironment would be very useful.

The paragraphs related to mitochondrial metabolism and other metabolic pathways (chapters 4 and 5, Table 4) are largely unrelated to M4-M5 AML and are poorly integrated in the review. Please simplify and reduce these sections by focusing on the concepts that are necessary to introduce the data focused on M4-M5 AML .

The same for paragraph 6.1: please provide a better focus on the monocytic differentiation role in venetoclax resistance.

Paragraph 6.2-6.3 are not connected to M4-M5 AML and should be deleted.

In chapter 7, the description of the biological/molecular/transcriptional profile of differentiated leukemic cells, as induced by targeted therapies should be discussed in comparison with pre-treatment M4-M5 AML cells: what do they have in common and what is different.

Comments on the Quality of English Language

English language is fine, few spelling mistakes should be checked.

Author Response

See enclosed coverletter.

Reviewer 2 Report

Comments and Suggestions for Authors

This is a very comprehensive, extensive work. It corresponds more to a text book chapter than to the usual format of a review.

The paper is very complete but this means that the specific link with the LMA FAB4/5 is sometimes weak. Several paragraphs give information not directly related to AML FAB4/5 and sometimes the conclusion of paragraphs is that the link with AML FAB4/5 is not known or that there is only one study not confirmed by others.

My suggestion would be that the text could be lightened somewhat to focus on the key, most significant elements related to FAB4/5 AML specificities and make the review easier and digestible for the reader to read.

As exemple

2.3. Expression of costimulatory molecules and checkpoint ligands by AML cells; high CD86 and 120 PD-L1 expression by FAB-M4/M5 AML cells

It is a long paragraph (interesting in general) but the conclusion is 

177 However, based on available studies there does not seem to be any strong associations between monocytic AML cell differentiation and (i) expression of costimulators/checkpoint molecules; or (ii) specific biological/therapeutic effects of checkpoint inhibition.

or

4.3. Mutations in the electron chain component genes; no association with FAB-M4/M5

or

7.1. IDH mutations and IDH inhibition; venetoclax sensitization and differentiation induction 898

IDH1 and IDH2 mutations are associated with the differentiated FAB-M2 phenotype

In conclusion very interesting work but for me, probably too long for one single review, a stricter focus on the subject could lighten the review and optimize the work

Author Response

See enclosed cover letter.

Reviewer 3 Report

Comments and Suggestions for Authors

Overall, authors need to carefully check the manuscript for misprints, although English language is fine.

Authors need to further highlight the relevance of the 2016 WHO classification that is mentioned several times throughout the paper.

One can understand the length of this paper as it is a review, however, it is extremely difficult to read the paper and go to the supplementary material and look for the information. This is unacceptable, the supplementary material should be included in the original manuscript as an appendix section, furthermore, each tome authors are making a reference to this table they should apply a proper marker so the reader may find the information instantly.

All tables need to be redesigned; all information gathered in them is difficult to understand due to poor organization. Authors need to be clever on how to properly show this information.

Authors have done extensive research concerning monocytic differentiation in acute myeloid leukemia cells. This manuscript has the potential  to be considered as “go to” on AML diagnostic criteria, biological heterogeneity, mitochondrial metabolism, resistance to and induction by targeted therapies, as it summarizes information regarding different kinds of studies (in vivo, in vitro, clinical), however, as it is right now is very tired to read.

Lastly, both as a writer and reviewer I am a visual person, hence I strongly advise authors to include diagrams or pictures to further complement the paper.

Comments on the Quality of English Language

Minor editing of English language required.

Author Response

See enclosed cover letter

Round 2

Reviewer 1 Report

Comments and Suggestions for Authors

The authors have addressed the reviewer's request and they have extensively modified the text. Figure 1 is very useful. The manuscript has been improved in focus and readability.